# Strategic Apple Tasting

**Keegan Harris**
Carnegie Mellon University
Pittsburgh, PA 15213
keeganh@cmu.edu

**Chara Podimata**
MIT & Archimedes/Athena RC
Cambridge, MA 02142
podimata@mit.edu

**Zhiwei Steven Wu**
Carnegie Mellon University
Pittsburgh, PA 15213
zstevenwu@cmu.edu

## Abstract

Algorithmic decision-making in high-stakes domains often involves assigning *decisions* to agents with *incentives* to strategically modify their input to the algorithm. In addition to dealing with incentives, in many domains of interest (e.g. lending and hiring) the decision-maker only observes feedback regarding their policy for rounds in which they assign a positive decision to the agent; this type of feedback is often referred to as *apple tasting* (or *one-sided*) feedback. We formalize this setting as an online learning problem with apple-tasting feedback where a *principal* makes decisions about a sequence of $T$ *agents*, each of which is represented by a *context* that may be strategically modified. Our goal is to achieve sublinear *strategic regret*, which compares the performance of the principal to that of the best fixed policy in hindsight, *if the agents were truthful when revealing their contexts*. Our main result is a learning algorithm which incurs $\tilde{\mathcal{O}}(\sqrt{T})$ strategic regret when the sequence of agents is chosen *stochastically*. We also give an algorithm capable of handling *adversarially-chosen* agents, albeit at the cost of $\tilde{\mathcal{O}}(T^{(d+1)/(d+2)})$ strategic regret (where $d$ is the dimension of the context). Our algorithms can be easily adapted to the setting where the principal receives *bandit* feedback—this setting generalizes both the linear contextual bandit problem (by considering agents with incentives) and the strategic classification problem (by allowing for partial feedback).

## 1 Introduction

Algorithmic systems have recently been used to aid in or automate decision-making in high-stakes domains (including lending and hiring) in order to, e.g., improve efficiency or reduce human bias [12, 1]. When subjugated to algorithmic decision-making in high-stakes settings, individuals have an incentive to *strategically* modify their observable attributes to appear more qualified. Such behavior is often observed in practice. For example, credit scores are often used to predict the likelihood an individual will pay back a loan on time if given one. Online articles with titles like *"9 Ways to Build and Improve Your Credit Fast"* are ubiquitous and offer advice such as "pay credit card balances strategically" in order to improve one's credit score with minimal effort [46]. In hiring, common advice ranges from curating a list of keywords to add to one's resume, to using white font in order to "trick" automated resume scanning software [23, 2]. If left unaccounted for, such strategic manipulations could result in individuals being awarded opportunities for which they are not qualified for, possibly at the expense of more deserving candidates. As a result, it is critical to keep individuals' incentives in mind when designing algorithms for learning and decision-making in high-stakes settings.

In addition to dealing with incentives, another challenge of designing learning algorithms for high-stakes settings is the possible *selection bias* introduced by the way decisions are made. In particular, decision-makers often only have access to feedback about the deployed policy from individuals that have received positive decisions (e.g., the applicant is given the loan, the candidate is hired to the job and then we can evaluate how good our decision was). In the language of online learning, this type of feedback is known as *apple tasting* (or *one-sided*) feedback. *When combined, these two complications (incentives & one-sided feedback) have the potential to amplify one other, as algorithms can learn*

*only when a positive decision is made, but individuals have an incentive to strategically modify their attributes in order to receive such positive decisions, which may interfere with the learning process.*

## 1.1 Contributions

We formalize our setting as a game between a *principal* and a sequence of $T$ *strategic agents*, each with an associated *context* $\mathbf{x}_t$ which describes the agent. At every time $t \in \{1, \dots, T\}$, the principal deploys a *policy* $\pi_t$, a mapping from contexts to binary *decisions* (e.g., whether to accept/reject a loan applicant). Given policy $\pi_t$, agent $t$ then presents a (possibly modified) context $\mathbf{x}'_t$ to the algorithm, and receives a decision $a_t = \pi_t(\mathbf{x}'_t)$. If $a_t = 1$, the principal observes *reward* $r_t(a_t) = r_t(1)$; if $a_t = 0$ they receive no feedback. $r_t(0)$ is assumed to be known and constant across rounds.[1] Our metric of interest is *strategic regret*, i.e., regret with respect to the best fixed policy in hindsight, *if agents were truthful when reporting their contexts*.

Our main result is an algorithm which achieves $\tilde{O}(\sqrt{T})$ strategic regret (with polynomial per-round runtime) when there is sufficient randomness in the distribution over agents (Algorithm 1). At a high level, our algorithm deploys a linear policy at every round which is appropriately shifted to account for the agents' strategic behavior. We identify a *sufficient* condition under which the data received by the algorithm at a given round is "clean", i.e. has not been strategically modified. Algorithm 1 then online-learns the relationship between contexts and rewards by only using data for which it is sure is clean.

In contrast to performance of algorithms which operate in the non-strategic setting, the regret of Algorithm 1 depends on an exponentially-large constant $c(d, \delta) \approx (1 - \delta)^{-d}$ due to the one-sided feedback available for learning, where $d$ is the context dimension and $\delta \in (0, 1)$ is a parameter which represents the agents' ability to manipulate. While this dependence on $c(d, \delta)$ is insignificant when the number of agents $T \to \infty$ (i.e. is very large), it may be problematic for the principal whenever $T$ is either small or unknown. To mitigate this issue, we show how to obtain $\tilde{O}(d \cdot T^{2/3})$ strategic regret by playing a modified version of the well-known *explore-then-commit* algorithm (Algorithm 2). At a high level, Algorithm 2 "explores" by always assigning action 1 for a fixed number of rounds (during which agents do not have an incentive to strategize) in order to collect sufficient information about the data-generating process. It then "exploits" by using this data learn a strategy-aware linear policy. Finally, we show how to combine Algorithm 1 and Algorithm 2 to achieve $\tilde{O}(\min\{c(d, \delta) \cdot \sqrt{T}, d \cdot T^{2/3}\})$ strategic regret whenever $T$ is unknown.

While the assumption of stochastically-chosen agents is well-motivated in general, it may be overly restrictive in some specific settings. Our next result is an algorithm which obtains $\tilde{O}(T^{(d+1)/(d+2)})$ strategic regret when agents are chosen *adversarially* (Algorithm 4). Algorithm 4 uses a variant of the popular Exp3 algorithm to trade off between a carefully constructed set of (exponentially-many) policies [6]. As a result, it achieves sublinear strategic regret when agents are chosen adversarially, but requires an exponentially-large amount of computation at every round.

Finally, we note that while our primary setting of interest is that of one-sided feedback, all of our algorithms can be easily extended to the more general setting in which the principal receives *bandit feedback* at each round, i.e. $r_t(0)$ is not constant and must be learned from data. To the best of our knowledge, we are the first to consider strategic learning in the contextual bandit setting.

## 1.2 Related work

**Strategic responses to algorithmic decision-making** There is a growing line of work at the intersection of economics and computation on algorithmic decision-making with incentives, under the umbrella of *strategic classification* or *strategic learning* [27, 3, 41, 22] focusing on online learning settings [21, 19], causal learning [52, 31, 29, 34], incentivizing desirable behavior [39, 28, 11, 42], incomplete information [30, 25, 37]. In its most basic form, a principal makes either a binary or real-valued prediction about a strategic agent, and receives *full feedback* (e.g., the agent's *label*) after the decision is made. While this setting is similar to ours, it crucially ignores the one-sided feedback structure present in many strategic settings of interest. In our running example of hiring, full feedback would correspond to a company not offering an applicant a job, and yet still getting to observe whether they would have been a good employee! As a result, such methods are not applicable in our setting. Concurrent work [18] studies the effects of bandit feedback in the related problem of *performative prediction* [47], which considers data distribution shifts at the *population level* in response to the deployment of a machine learning model. In contrast, our focus is on strategic responses to machine

---

[1]We relax this assumption at later parts of the paper with virtually no impact on our results.

learning models at the *individual level* under apple tasting and bandit feedback. Ahmadi et al. [4] study an online strategic learning problem in which they consider "bandit feedback" *with respect to the deployed classifier*. In contrast, we use the term "bandit feedback" to refer to the fact that we only see the outcome when for the action/decision taken.

**Apple tasting and online learning** Helmbold et al. [32] introduce the notion of apple-tasting feedback for online learning. In particular, they study a binary prediction task over "instances" (e.g., fresh/rotten apples), in which a positive prediction is interpreted as accepting the instance (i.e. "tasting the apple") and a negative prediction is interpreted as rejecting the instance (i.e., *not* tasting the apple). The learner only gets feedback when the instance is accepted (i.e., the apple is tasted). While we are the first to consider classification under incentives with apple tasting feedback, similar feedback models have been studied in the context of algorithmic fairness [9], partial-monitoring games [5], and recidivism prediction [24]. A related model of feedback is that of *contaminated controls* [40], which considers learning from (1) a treated group which contains only *treated* members of the agent population and (2) a "contaminated" control group with samples from the *entire* agent population (not just those under *control*). Technically, our results are also related to a line of work in contextual bandits which shows that greedy algorithms without explicit exploration can achieve sublinear regret as long as the underlying context distribution is sufficiently diverse [49, 8, 38, 53, 48].

**Bandits and agents** A complementary line of work to ours is that of *Bayesian incentive-compatible* (BIC) exploration in multi-armed bandit problems [43, 35, 50, 36, 44, 45]. Under such settings, the goal of the principal is to *persuade* a sequence of $T$ agents with incentives to explore across several different actions with bandit feedback. In contrast, in our setting it is the principal, not the agents, who is the one taking actions with partial feedback. As a result there is no need for persuasion, but the agents now have an incentive to strategically modify their behavior in order to receive a more desirable decision/action.

**Other related work** Finally, our work is broadly related to the literature on learning in repeated Stackelberg games [7, 55], online Bayesian persuasion [16, 17, 13], and online learning in principal-agent problems [20, 33, 56]. In the repeated Stackelberg game setting, the principal (leader) commits to a mixed strategy over a finite set of actions, and the agent (follower) best-responds by playing an action from a finite set of best-responses. Unlike in our setting, both the principal's and agent's payoffs can be represented by matrices. In contrast, in our setting the principal commits to a pure strategy from a continuous set of actions, and the agent best-responds by playing an action from a continuous set. In online Bayesian persuasion, the principal (sender) commits to a "signaling policy" (a random mapping from "states of the world" to receiver actions) and the agent (receiver) performs a posterior update on the state based on the principal's signal, then takes an action from a (usually finite) set. In both this setting and ours, the principal's action is a policy. However in our setting the policy is a linear decision rule, whereas in the Bayesian persuasion setting, the policy is a set of conditional probabilities which form an "incentive compatible" signaling policy. This difference in the policy space for the principal typically leads to different algorithmic ideas being used in the two settings. Strategic learning problems are, broadly speaking, instances of principal-agent problems. In contract design, the principal commits to a contract (a mapping from "outcomes" to agent payoffs). The agent then takes an action, which affects the outcome. In particular, they take the action which maximizes their expected payoff, subject to some cost of taking the action. The goal of the principal is to design a contract such that their own expected payoff is maximized. While the settings are indeed similar, there are several key differences. First, in online contract design the principal always observes the outcome, whereas in our setting the principal only observes the reward if a positive decision is made. Second, the form of the agent's best response is different, which leads to different agent behavior and, as a result, different online algorithms for the principal.

## 2   Setting and background

We consider a game between a *principal* and a sequence of $T$ *agents*. Each agent is associated with a *context* $\mathbf{x}_t \in \mathcal{X} \subseteq \mathbb{R}^d$, which characterizes their attributes (e.g., a loan applicant's credit history/report). At time $t$, the principal commits to a *policy* $\pi_t : \mathcal{X} \rightarrow \{1, 0\}$, which maps from contexts to binary *decisions* (e.g., whether to accept/reject the loan application). We use $a_t = 1$ to denote the the principal's positive decision at round $t$ (e.g., agent $t$'s loan application is approved), and $a_t = 0$ to denote a negative decision (e.g., the loan application is rejected). Given $\pi_t$, agent $t$ *best-responds* by strategically modifying their context within their *effort budget* as follows:

**Definition 2.1** (Agent best response; lazy tiebreaking). *Agent $t$ best-responds to policy $\pi_t$ by modifying their context according to the following optimization program.*

$$\mathbf{x}'_t \in \arg\max_{\mathbf{x}' \in \mathcal{X}} \; \mathbb{1}\{\pi_t(\mathbf{x}') = 1\}$$
$$s.t. \; \|\mathbf{x}' - \mathbf{x}_t\|_2 \leq \delta$$

*Furthermore, we assume that if an agent is indifferent between two (modified) contexts, they choose the one which requires the least amount of effort to obtain (i.e., agents are* lazy *when tiebreaking).*

In other words, every agent wants to receive a positive decision, but has only a limited ability to modify their (initial) context (represented by $\ell_2$ budget $\delta$).[2] Such an effort budget may be induced by time or monetary constraints and is a ubiquitous model of agent behavior in the strategic learning literature (e.g., [39, 28, 19, 10]). We focus on *linear thresholding policies* where the principal assigns action $\pi(\mathbf{x}') = 1$, if and only if $\langle \boldsymbol{\beta}, \mathbf{x}' \rangle \geq \gamma$ for some $\boldsymbol{\beta} \in \mathbb{R}^d$, $\gamma \in \mathbb{R}$. We refer to $\langle \boldsymbol{\beta}, \mathbf{x}'_t \rangle = \gamma$ as the *decision boundary*. For linear thresholding policies, the agent's best-response according to Definition 2.1 is to modify their context in the direction of $\boldsymbol{\beta}/\|\boldsymbol{\beta}\|_2$ until the decision-boundary is reached (if it can indeed be reached). While we present our results for *lazy tiebreaking* for ease of exposition, all of our results can be readily extended to the setting in which agents best-respond with a "trembling hand", i.e. *trembling hand tiebreaking*. Under this setting, we allow agents who strategically modify their contexts to "overshoot" the decision boundary by some bounded amount, which can be either stochastic or adversarially-chosen. See Appendix D for more details.

The principal observes $\mathbf{x}'_t$ and plays action $a_t = \pi_t(\mathbf{x}'_t)$ according to policy $\pi_t$. If $a_t = 0$, the principal receives some known, *constant* reward $r_t(0) := r_0 \in \mathbb{R}$. On the other hand, if the principal assigns action $a_t = 1$, we assume that the reward the principal receives is linear in the agent's *unmodified* context, i.e.,

$$r_t(1) := \langle \boldsymbol{\theta}^{(1)}, \mathbf{x}_t \rangle + \epsilon_t \tag{1}$$

for some *unknown* $\boldsymbol{\theta}^{(1)} \in \mathbb{R}^d$, where $\epsilon_t$ is i.i.d. zero-mean sub-Gaussian random noise with (known) variance $\sigma^2$. Note that $r_t(1)$ is observed *only* when the principal assigns action $a_t = 1$, and *not* when $a_t = 0$. Following Helmbold et al. [32], we refer to such feedback as *apple tasting* (or *one-sided*) feedback. Mapping to our lending example, the reward a bank receives for rejecting a particular loan applicant is the same across all applicants, whereas their reward for a positive decision could be anywhere between a large, negative reward (e.g., if a loan is never repaid) to a large, positive reward (e.g., if the loan is repaid on time, with interest).

The most natural measure of performance in our setting is that of *Stackelberg regret*, which compares the principal's reward over $T$ rounds with that of the optimal policy *given that agents strategize*.

**Definition 2.2** (Stackelberg regret). *The Stackelberg regret of a sequence of policies $\{\pi_t\}_{t \in [T]}$ on agents $\{\mathbf{x}_t\}_{t \in [T]}$ is*

$$\mathrm{Reg}_{\mathtt{Stackel}}(T) := \sum_{t \in [T]} r_t(\tilde{\pi}^*(\tilde{\mathbf{x}}_t)) - \sum_{t \in [T]} r_t(\pi_t(\mathbf{x}'_t))$$

*where $\tilde{\mathbf{x}}_t$ is the best-response from agent $t$ to policy $\tilde{\pi}^*$ and $\tilde{\pi}^*$ is the optimal-in-hindsight policy, given that agents best-respond according to Definition 2.1.*

A stronger measure of performance is that of *strategic regret*, which compares the principal's reward over $T$ rounds with that of the optimal policy *had agents reported their contexts truthfully*.

**Definition 2.3** (Strategic regret). *The strategic regret of a sequence of policies $\{\pi_t\}_{t \in [T]}$ on agents $\{\mathbf{x}_t\}_{t \in [T]}$ is*

$$\mathrm{Reg}_{\mathtt{strat}}(T) := \sum_{t \in [T]} r_t(\pi^*(\mathbf{x}_t)) - \sum_{t \in [T]} r_t(\pi_t(\mathbf{x}'_t))$$

*where $\pi^*(\mathbf{x}_t) = 1$ if $\langle \boldsymbol{\theta}^{(1)}, \mathbf{x}_t \rangle \geq r_0$ and $\pi^*(\mathbf{x}_t) = 0$ otherwise.*

---

[2] Our results readily extend to the setting in which the agent's effort constraint takes the form of an ellipse rather than a sphere. Under this setting, the agent effort budget constraint in Definition 2.1 would be $\|A^{1/2}(\mathbf{x}' - \mathbf{x}_t)\|_2 \leq \delta$, where $A \in \mathbb{R}^{d \times d}$ is some positive definite matrix. If $A$ is known to the principal, this may just be viewed as a linear change in the feature representation.

For $t = 1, \ldots, T$:
1. Principal publicly commits to a mapping $\pi_t : \mathcal{X} \to \{1, 0\}$.
2. Agent $t$ arrives with context $\mathbf{x}_t \in \mathcal{X}$ (hidden from the principal).
3. Agent $t$ strategically modifies context from $\mathbf{x}_t$ to $\mathbf{x}_t'$ according to Definition 2.1.
4. Principal observes (modified) context $\mathbf{x}_t'$ and plays action $a_t = \pi_t(\mathbf{x}_t')$.
5. Principal observes $r_t(1) := \langle \boldsymbol{\theta}^{(1)}, \mathbf{x}_t \rangle + \epsilon_t$, if and only if $a_t = 1$.

Figure 1: Summary of our model.

**Proposition 2.4.** *Strategic regret is a* stronger *performance notion compared to Stackelberg regret, i.e.,* $\mathrm{Reg}_{\mathtt{Stackel}}(T) \leq \mathrm{Reg}_{\mathtt{strat}}(T)$.

*Proof.* The proof follows from the corresponding regret definitions and the fact that the principal's reward is determined by the original (unmodified) agent contexts.

$$
\begin{aligned}
R_{\mathtt{Stackel}}(T) &:= \sum_{t \in [T]} r_t(\tilde{\pi}^*(\tilde{\mathbf{x}}_t)) - \sum_{t \in [T]} r_t(\pi_t(\mathbf{x}_t')) \\
&= \sum_{t \in [T]} r_t(\tilde{\pi}^*(\tilde{\mathbf{x}}_t)) - \sum_{t \in [T]} r_t(\pi^*(\mathbf{x}_t)) + \sum_{t \in [T]} r_t(\pi^*(\mathbf{x}_t)) - \sum_{t \in [T]} r_t(\pi_t(\mathbf{x}_t')) \\
&\leq 0 + R_{\mathtt{strat}}(T)
\end{aligned}
$$

where the last line follows from the fact that the principal's reward from the optimal policy when the agent strategizes is at most their optimal reward when agents do not strategize. $\square$

Because of Proposition 2.4, we focus on strategic regret, and use the shorthand $\mathrm{Reg}_{\mathtt{strat}}(T) = \mathrm{Reg}(T)$ for the remainder of the paper. Strategic regret is a strong notion of optimality, as we are comparing the principal's performance with that of the optimal policy for an easier setting, in which agents do not strategize. Moreover, the apple tasting feedback introduces additional challenges which require new algorithmic ideas to solve, since the principal needs to assign actions to both (1) learn about $\boldsymbol{\theta}^{(1)}$ (which can only be done when action 1 is assigned) and (2) maximize rewards in order to achieve sublinear strategic regret. See Figure 1 for a summary of the setting we consider.

We conclude this section by pointing out that our results also apply to the more challenging setting of *bandit feedback*, in which $r_t(1)$ is defined as in Equation (1), $r_t(0) := \langle \boldsymbol{\theta}^{(0)}, \mathbf{x}_t \rangle + \epsilon_t$ and only $r_t(a_t)$ is observed at each time-step. We choose to highlight our results for apple tasting feedback since this is the type of feedback received by the principal in our motivating examples. Finally, we note that $\widetilde{\mathcal{O}}(\cdot)$ hides polylogarithmic factors, and that all proofs can be found in the Appendix.

## 3 Strategic classification with apple tasting feedback

In this section, we present our main results: provable guarantees for online classification of strategic agents under apple tasting feedback. Our results rely on the following assumption.

**Assumption 3.1** (Bounded density ratio). *Let* $f_{U^d} : \mathcal{X} \to \mathbb{R}_{\geq 0}$ *denote the density function of the uniform distribution over the $d$-dimensional unit sphere. We assume that agent contexts $\{\mathbf{x}_t\}_{t \in [T]}$ are drawn i.i.d. from a distribution over the $d$-dimensional unit sphere with density function $f : \mathcal{X} \to \mathbb{R}_{\geq 0}$ such that $\frac{f(\mathbf{x})}{f_{U^d}(\mathbf{x})} \geq c_0 > 0$, $\forall \mathbf{x} \in \mathcal{X}$.*[3]

Assumption 3.1 is a condition on the *initial* agent contexts $\{\mathbf{x}_t\}_{t \in [T]}$, *before* they are strategically modified. Indeed, one would expect the distribution over *modified* agent contexts to be highly discontinuous in a way that depends on the sequence of policies deployed by the principal. Furthermore, none of our algorithms need to know the value of $c_0$. As we will see in the sequel, this assumption allows us to handle apple tasting feedback by *relying on the inherent diversity in the agent population*

---

[3]Our restriction to the *unit* sphere is without loss of generality. All of our results and analysis extend readily to the setting where contexts are drawn from a distribution over the $d$-dimensional sphere with radius $R > 0$.

**ALGORITHM 1:** Strategy-Aware OLS with Apple Tasting Feedback (`SA-OLS`)

---

Assign action 1 for the first $d$ rounds.

Set $\mathcal{D}_{d+1} = \{(\mathbf{x}_s, r_s^{(1)})\}_{s=1}^d$.

**for** $t = d + 1, \ldots, T$ **do**

    Estimate $\boldsymbol{\theta}^{(1)}$ as $\widehat{\boldsymbol{\theta}}_t^{(1)}$ using OLS and data $\mathcal{D}_t$.

    Assign action $a_t = 1$ if $\langle \widehat{\boldsymbol{\theta}}_t^{(1)}, \mathbf{x}_t' \rangle \geq \delta \cdot \|\widehat{\boldsymbol{\theta}}_t^{(1)}\|_2 + r_0$.

    **if** $\langle \widehat{\boldsymbol{\theta}}_t^{(1)}, \mathbf{x}_t' \rangle > \delta \|\widehat{\boldsymbol{\theta}}_t^{(1)}\|_2 + r_0$ **then**

        Conclude that $\mathbf{x}_t' = \mathbf{x}_t$.

        $\mathcal{D}_{t+1} = \mathcal{D}_t \cup \{(\mathbf{x}_t, r_t^{(1)})\}$

    **else**

        $\mathcal{D}_{t+1} = \mathcal{D}_t$

---

*for exploration*; a growing area of interest in the online learning literature (see references in Section 1.2). Moreover, such assumptions often hold in practice. For example, in the related problem of (non-strategic) contextual bandits (we will later show how our results extend to the strategic version of this problem), Bietti et al. [14] find that a greedy algorithm with no explicit exploration achieved the second-best empirical performance across a large number of datasets when compared to many popular contextual bandit algorithms. In our settings of interest (e.g. lending, hiring), such an assumption is reasonable if there is sufficient diversity in the applicant pool. In Section 4 we show how to remove this assumption, albeit at the cost of worse regret rates and exponential computational complexity.

At a high level, our algorithm (formally stated in Algorithm 1) relies on three key ingredients to achieve sublinear strategic regret:

1. A running estimate of $\boldsymbol{\theta}^{(1)}$ is used to compute a linear policy, which separates agents who receive action 1 from those who receive action 0. Before deploying, we shift the decision boundary by the effort budget $\delta$ to account for the agents strategizing.

2. We maintain an estimate of $\boldsymbol{\theta}^{(1)}$ (denoted by $\widehat{\boldsymbol{\theta}}^{(1)}$) and only updating it when $a_t = 1$ and we can ensure that $\mathbf{x}_t' = \mathbf{x}_t$.

3. We assign actions "greedily" (i.e. using no explicit exploration) w.r.t. the shifted linear policy.

**Shifted linear policy** If agents were *not* strategic, assigning action 1 if $\langle \widehat{\boldsymbol{\theta}}_t^{(1)}, \mathbf{x}_t \rangle \geq r_0$ and action 0 otherwise would be a reasonable strategy to deploy, given that $\widehat{\boldsymbol{\theta}}_t^{(1)}$ is our "best estimate" of $\boldsymbol{\theta}^{(1)}$ so far. Recall that the strategically modified context $\mathbf{x}_t'$ is s.t., $\|\mathbf{x}_t' - \mathbf{x}_t\| \leq \delta$. Hence, in Algorithm 1, we shift the linear policy by $\delta \|\widehat{\boldsymbol{\theta}}^{(1)}\|_2$ to account for strategically modified contexts. Now, action 1 is only assigned if $\langle \widehat{\boldsymbol{\theta}}_t^{(1)}, \mathbf{x}_t \rangle \geq \delta \|\widehat{\boldsymbol{\theta}}^{(1)}\|_2 + r_0$. This serves two purposes: (1) It makes it so that any agent with unmodified context $\mathbf{x}$ such that $\langle \widehat{\boldsymbol{\theta}}_t^{(1)}, \mathbf{x} \rangle < r_0$ cannot receive action 1, no matter how they strategize. (2) It forces some agents with contexts in the band $r_0 \leq \langle \widehat{\boldsymbol{\theta}}_t^{(1)}, \mathbf{x} \rangle < \delta \|\widehat{\boldsymbol{\theta}}^{(1)}\|_2 + r_0$ to strategize in order to receive action 1. **Estimating $\boldsymbol{\theta}^{(1)}$** After playing action 1 for the first $d$ rounds, Algorithm 1 forms an initial estimate of $\boldsymbol{\theta}^{(1)}$ via ordinary least squares (OLS). Note that since the first $d$ agents will receive action 1 regardless of their context, they have no incentive to modify and thus $\mathbf{x}_t' = \mathbf{x}_t$ for $t \leq d$. In future rounds, the algorithm's estimate of $\boldsymbol{\theta}^{(1)}$ is only updated whenever $\mathbf{x}_t'$ lies *strictly* on the positive side of the linear decision boundary. We call these contexts *clean*, and can infer that $\mathbf{x}_t' = \mathbf{x}_t$ due to the lazy tiebreaking assumption in Definition 2.1 (i.e. agents will not strategize more than is necessary to receive the positive classification).

**Condition 3.2** (Sufficient condition for $\mathbf{x}' = \mathbf{x}$). *Given a shifted linear policy parameterized by $\boldsymbol{\beta}^{(1)} \in \mathbb{R}^d$, we say that a context $\mathbf{x}'$ is* clean *if* $\langle \boldsymbol{\beta}^{(1)}, \mathbf{x}' \rangle > \delta \|\boldsymbol{\beta}^{(1)}\|_2 + r_0$.

**Greedy action assignment** By assigning actions greedily according to the current (shifted) linear policy, we are relying on the diversity in the agent population for implicit exploration (i.e., to collect more datapoints to update our estimate of $\boldsymbol{\theta}^{(1)}$). As we will show, this implicit exploration

is sufficient to achieve $\widetilde{\mathcal{O}}(\sqrt{T})$ strategic regret under Assumption 3.1, albeit at the cost of an exponentially-large (in $d$) constant which depends on the agents' ability to manipulate ($\delta$).

We are now ready to present our main result: strategic regret guarantees for Algorithm 1 under apple tasting feedback.

**Theorem 3.3** (Informal; detailed version in Theorem B.1). *With probability $1 - \gamma$, Algorithm 1 achieves the following performance guarantee:*

$$\mathrm{Reg}(T) \leq \widetilde{\mathcal{O}}\left(\frac{1}{c_0 \cdot c_1(d,\delta) \cdot c_2(d,\delta)}\sqrt{d\sigma^2 T \log(4dT/\gamma)}\right)$$

*where $c_0$ is a lower bound on the density ratio as defined in Assumption 3.1, $c_1(d,\delta) := \mathbb{P}_{\mathbf{x}\sim U^d}(\mathbf{x}[1] \geq \delta) \geq \Theta\left(\frac{(1-\delta)^{d/2}}{d^2}\right)$ for sufficiently large $d$ and $c_2(d,\delta) := \mathbb{E}_{\mathbf{x}\sim U^d}[\mathbf{x}[2]^2|\mathbf{x}[1] \geq \delta] \geq \left(\frac{3}{4} - \frac{1}{2}\delta - \frac{1}{4}\delta^2\right)^3$, where $\mathbf{x}[i]$ denotes the $i$-th coordinate of a vector $\mathbf{x}$.[4]*

*Proof sketch.* Our analysis begins by using properties of the strategic agents and shifted linear decision boundary to upper-bound the per-round strategic regret for rounds $t > d$ by a term proportional to $\|\widehat{\boldsymbol{\theta}}_t^{(1)} - \boldsymbol{\theta}^{(1)}\|_2$, i.e., our instantaneous estimation error for $\boldsymbol{\theta}^{(1)}$. Next we show that

$$\|\widehat{\boldsymbol{\theta}}_t^{(1)} - \boldsymbol{\theta}^{(1)}\|_2 \leq \frac{\left\|\sum_{s=1}^t \mathbf{x}_s \epsilon_s \mathbb{1}\{\mathcal{I}_s^{(1)}\}\right\|_2}{\lambda_{min}(\sum_{s=1}^t \mathbf{x}_s\mathbf{x}_s^\top \mathbb{1}\{\mathcal{I}_s^{(1)}\})}$$

where $\lambda_{min}(M)$ is the minimum eigenvalue of (symmetric) matrix $M$, and $\mathcal{I}_s^{(1)} = \{\langle\widehat{\boldsymbol{\theta}}_s^{(1)}, \mathbf{x}_s\rangle \geq \delta\|\widehat{\boldsymbol{\theta}}_s^{(1)}\|_2 + r_0\}$ is the event that Algorithm 1 assigns action $a_s = 1$ and can verify that $\mathbf{x}'_s = \mathbf{x}_s$. We upper-bound the numerator using a variant of Azuma's inequality for martingales with subgaussian tails. Next, we use properties of Hermitian matrices to show that $\lambda_{min}(\sum_{s=1}^t \mathbf{x}_s\mathbf{x}_s^\top \mathbb{1}\{\mathcal{I}_s^{(1)}\})$ is lower-bounded by two terms: one which may be bounded w.h.p. by using the extension of Azuma's inequality for matrices, and one of the form $\sum_{s=1}^t \lambda_{min}(\mathbb{E}_{s-1}[\mathbf{x}_s\mathbf{x}_s^\top \mathbb{1}\{\mathcal{I}_s^{(1)}\}])$, where $\mathbb{E}_{s-1}$ denotes the expected value conditioned on the filtration up to time $s$. Note that up until this point, we have only used the fact that contexts are drawn i.i.d. from a *bounded* distribution.

Using Assumption 3.1 on the bounded density ratio, we can lower bound $\lambda_{min}(\mathbb{E}_{s-1}[\mathbf{x}_s\mathbf{x}_s^\top \mathbb{1}\{\mathcal{I}_s^{(1)}\}])$ by $\lambda_{min}(\mathbb{E}_{U^d,s-1}[\mathbf{x}_s\mathbf{x}_s^\top \mathbb{1}\{\mathcal{I}_s^{(1)}\}])$, *where the expectation is taken with respect to the uniform distribution over the $d$-dimensional ball*. We then use properties of the uniform distribution to show that $\lambda_{min}(\mathbb{E}_{U^d,s-1}[\mathbf{x}_s\mathbf{x}_s^\top \mathbb{1}\{\mathcal{I}_s^{(1)}\}]) \geq \mathcal{O}(c_0 \cdot c(d,\delta))$. Putting everything together, we get that $\|\widehat{\boldsymbol{\theta}}_t^{(1)} - \boldsymbol{\theta}^{(1)}\|_2 \leq (c_0 \cdot c(d,\delta) \cdot \sqrt{t})^{-1}$ with high probability. Via a union bound and the fact that $\sum_{t\in[T]} \frac{1}{\sqrt{t}} \leq 2T$, we get that $\mathrm{Reg}(T) \leq \widetilde{\mathcal{O}}(\frac{1}{c_0 \cdot c(d,\delta)}\sqrt{T})$. Finally, we use tools from high-dimensional geometry to lower bound the volume of a spherical cap and we show that for sufficiently large $d$, $c_1(d,\delta) \geq \Theta\left(\frac{(1-\delta)^{d/2}}{d^2}\right)$. □

## 3.1 High-dimensional contexts

While we typically think of the number of agents $T$ as growing and the context dimension $d$ as constant in our applications of interest, there may be situations in which $T$ is either unknown or small. Under such settings, the $1/c(d,\delta)$ dependence in the regret bound (where $c(d,\delta) = c_1(d,\delta) \cdot c_2(d,\delta)$) may become problematic if $\delta$ is close to 1. This begs the question: "Why restrict the OLS estimator in Algorithm 1 to use only clean contexts (as defined in Condition 3.2)?" Perhaps unsurprisingly, we show in Appendix B that the estimate $\widehat{\boldsymbol{\theta}}^{(1)}$ given by OLS will be inconsistent if even a constant fraction of agents strategically modify their contexts.

---

[4]While we assume that $\delta$ is known to the principal, Algorithm 1 is fairly robust to overestimates of $\delta$, in the sense that (1) it will still produce a consistent estimate for $\boldsymbol{\theta}^{(1)}$ (albeit at a rate which depends on the overestimate instead of the actual value of $\delta$) and (2) it will incur a constant penalty in regret which is proportional to the amount of over-estimation.

**ALGORITHM 2:** Explore-Then-Commit

---

**Input :** Time horizon $T$, failure probability $\gamma$

Set $T_0$ according to Theorem B.9

Assign action 1 for the first $T_0$ rounds

Estimate $\boldsymbol{\theta}^{(1)}$ as $\hat{\boldsymbol{\theta}}_{T_0}^{(1)}$ via OLS

**for** $t = T_0 + 1, \ldots, T$ **do**

    Assign action $a_t = 1$ if $\langle \hat{\boldsymbol{\theta}}_{T_0}^{(1)}, \mathbf{x}_t \rangle \geq \delta \cdot \|\hat{\boldsymbol{\theta}}_{T_0}^{(1)}\|_2$ and action $a_t = 0$ otherwise

---

Given the above, it seems reasonable to restrict ourselves to learning procedures which only use data from agents for which the principal can be sure that $\mathbf{x}' = \mathbf{x}$. Under such a restriction, it is natural to ask whether there exists some sequence of linear polices which maximizes the number of points of the form $(\mathbf{x}'_t, r_t(1))$ for which the principal can be sure that $\mathbf{x}'_t = \mathbf{x}_t$. Again, the answer is no:

**Proposition 3.4.** *For any sequence of linear policies $\{\boldsymbol{\beta}_t\}_t$, the expected number of clean points is:*

$$\mathbb{E}_{\mathbf{x}_1, \ldots, \mathbf{x}_T \sim U^d} \left[ \sum_{t \in [T]} \mathbb{1}\{\langle \mathbf{x}_t, \boldsymbol{\beta}_t \rangle > \delta \|\boldsymbol{\beta}_t\|_2\} \right] = c_1(d, \delta) \cdot T$$

*when (initial) contexts are drawn uniformly from the $d$-dimensional unit sphere.*

The proof follows from the rotational invariance of the uniform distribution over the unit sphere. Intuitively, Proposition 3.4 implies that any algorithm which wishes to learn $\boldsymbol{\theta}^{(1)}$ using clean samples will only have $c_1(d, \delta) \cdot T$ datapoints in expectation. Observe that this dependence on $c_1(d, \delta)$ arises as a direct result of the agents' ability to strategize. We remark that a similar constant often appears in the regret analysis of BIC bandit algorithms (see Section 1.2). Much like our work, [43] find that their regret rates depend on a constant which may be arbitrarily large, depending on how hard it is to persuade agents to take the principal's desired action in their setting. The authors conjecture that this dependence is an inevitable "price of incentive-compatibility". While our results do not rule out better strategic regret rates in $d$ for more complicated algorithms (e.g., those which deploy non-linear policies), it is often unclear how strategic agents would behave in such settings, both in theory (Definition 2.1 would require agents to solve a non-convex optimization with potentially no closed-form solution) and in practice, making the analysis of such nonlinear policies difficult in strategic settings.

We conclude this section by showing that polynomial dependence on $d$ is possible, at the cost of $\widetilde{\mathcal{O}}(T^{2/3})$ strategic regret. Specifically, we provide an algorithm (Algorithm 3) which obtains the following regret guarantee whenever $T$ is small or unknown, which uses Algorithm 1 and a variant of the explore-then-commit algorithm (Algorithm 2) as subroutines:

**Theorem 3.5** (Informal; details in Theorem B.13). *Algorithm 3 incurs expected strategic regret*

$$\mathbb{E}[\text{Reg}(T)] = \widetilde{\mathcal{O}}\left( \min\left\{ \frac{d^{5/2}}{(1-\delta)^{d/2}} \cdot \sqrt{T}, d \cdot T^{2/3} \right\} \right),$$

*where the expectation is taken with respect to the sequence of contexts $\{\mathbf{x}_t\}_{t \in [T]}$ and random noise $\{\epsilon_t\}_{t \in [T]}$.*

The algorithm proceeds by playing a "strategy-aware" variant of explore-then-commit (Algorithm 2) with a doubling trick until the switching time $\tau^* = g(d, \delta)$ is reached. Note that $g(d, \delta)$ is a function of both $d$ and $\delta$, *not* $c_0$. If round $\tau^*$ is indeed reached, the algorithm switches over to Algorithm 1 for the remaining rounds.

**Extension to bandit feedback** Algorithm 1 can be extended to handle bandit feedback by explicitly keeping track of an estimate $\widehat{\boldsymbol{\theta}}^{(0)}$ of $\boldsymbol{\theta}^{(0)}$ via OLS, assigning action $a_t = 1$ if and only if $\langle \widehat{\boldsymbol{\theta}}_t^{(1)} - \widehat{\boldsymbol{\theta}}_t^{(0)}, \mathbf{x}'_t \rangle \geq \delta \cdot \|\widehat{\boldsymbol{\theta}}_t^{(1)} - \widehat{\boldsymbol{\theta}}_t^{(0)}\|_2$, and updating the OLS estimate of $\widehat{\boldsymbol{\theta}}^{(0)}$ whenever $a_t = 0$ (since agents will not strategize to receive action 0). Algorithm 3 may be extended to bandit feedback by "exploring" for twice as long in Algorithm 2, in addition to using the above modifications. In both cases, the strategic regret rates are within a constant factor of the rates obtained in Theorem 3.3 and Theorem 3.5.

**ALGORITHM 3:** Strategy-aware online classification with unknown time horizon

---

Compute switching time $\tau^* = g(d, \delta)$
Let $\tau_0 = 1$
**for** $i = 1, 2, 3, \ldots$ **do**
    Let $\tau_i = 2 \cdot \tau_{i-1}$
    **if** $\sum_{j=1}^{i} \tau_j < \tau^*$ **then**
        Run Algorithm 2 with time horizon $\tau_i$ and failure probability $1/\tau_i^2$
    **else**
        Break and run Algorithm 1 for the remainder of the rounds

---

## 4   Beyond stochastic contexts

In this section, we allow the sequence of initial agent contexts to be chosen by an (oblivious) *adversary*. This requires new algorithmic ideas, as the regression-based algorithms of Section 3 suffer *linear* strategic regret under this adversarial setting. Our algorithm (Algorithm 4) is based on the popular EXP3 algorithm [6]. At a high level, Algorithm 4 maintains a probability distribution over "experts", i.e., a discretized grid $\mathcal{E}$ over carefully-selected policies. In particular, each grid point $\mathbf{e} \in \mathcal{E} \subseteq \mathbb{R}^d$ represents an "estimate" of $\boldsymbol{\theta}^{(1)}$, and corresponds to a slope vector which parameterizes a (shifted) linear threshold policy, like the ones considered in Section 3. We use $a_{t,\mathbf{e}}$ to refer to the action played by the principal at time $t$, had they used the linear threshold policy parameterized by expert $\mathbf{e}$. At every time-step, (1) the adversary chooses an agent $\mathbf{x}_t$, (2) a slope vector $\mathbf{e}_t \in \mathcal{E}$ is selected according to the current distribution, (3) the principal commits to assigning action 1 if and only if $\langle \mathbf{e}_t, \mathbf{x}'_t \rangle \geq \delta \|\mathbf{e}_t\|_2$, (4) the agent strategically modifies their context $\mathbf{x}_t \to \mathbf{x}'_t$, and (5) the principal assigns an action $a_t$ according to the policy and receives the associated reward $r_t(a_t)$ (under apple tasting feedback).

Algorithm EXP4, which maintains a distribution over experts and updates the loss of *all* experts based on the current action taken, is not directly applicable in our setting as the strategic behavior of the agents prevents us from inferring the loss of each expert at every time-step [**?** ]. This is because if $\mathbf{x}'_t \neq \mathbf{x}_t$ under the thresholding policy associated with expert $\mathbf{e}$), it is generally not possible to "back out" $\mathbf{x}_t$ given $\mathbf{x}'_t$, which prevents us from predicting the counterfactual context the agent would have modified to had the principal been using expert $\mathbf{e}'$ instead. As a result, we use a modification of the standard importance-weighted loss estimator to update the loss of *only the policy played by the algorithm* (and therefore the distribution over policies). Our regret guarantees for Algorithm 4 are as follows:

**Theorem 4.1** (Informal; detailed version in Theorem C.1). *Algorithm 4 incurs expected strategic regret* $\mathbb{E}[\text{Reg}(T)] = \widetilde{\mathcal{O}}(T^{(d+1)/(d+2)})$.

We remark that Algorithm 4 may be extended to handle settings in which agents are selected by an *adaptive* adversary by using EXP3.P [6] in place of EXP3.

*Proof sketch.* The analysis is broken down into two parts. In the first part, we bound the regret w.r.t. the best policy on the grid. In the second, we bound the error incurred for playing policies on the grid, rather than the continuous space of policies. We refer to this error as the *Strategic Discretization Error* $(SDE(T))$. The analysis of the regret on the grid mostly follows similar steps to the analysis of EXP3 / EXP4. The important difference is that we shift the reward obtained by $a_t$, by a factor of $1 + \lambda$, where $\lambda$ is a (tunable) parameter of the algorithm. This shifting (which does not affect the regret, since all the losses are shifted by the same fixed amount) guarantees that the losses at each round are non-negative and bounded with high probability. Technically, this requires bounding the tails of the subgaussian of the noise parameters $\epsilon_t$.

We now shift our attention to bounding $SDE(T)$. The standard analysis of the discretization error in the non-strategic setting does not go through for our setting, since an agent may strategize very differently with respect to two policies which are "close together" in $\ell_2$ distance, depending on the agent's initial context. Our analysis proceeds with a case-by-case basis. Consider the best expert $\mathbf{e}^*$ in the grid. If $a_{t,\mathbf{e}^*} = \pi^*(\mathbf{x}_t)$ (i.e., the action of the best expert matches that of the optimal policy), there is no discretization error in round $t$. Otherwise, if $a_{t,\mathbf{e}^*} \neq \pi^*(\mathbf{x}_t)$, we show that the per-round $SDE$ is upper-bounded by a term which looks like twice the discretization upper-bound for the non-strategic setting, plus an additional term. We show that this additional term must always be non-positive by considering two subcases ($a_{t,\mathbf{e}^*} = 1, \pi^*(\mathbf{x}_t) = 0$ and $a_{t,\mathbf{e}^*} = 0, \pi^*(\mathbf{x}_t) = 1$) and using properties about how agents strategize against the deployed algorithmic policies.   □

**ALGORITHM 4:** EXP3 with strategy-aware experts (EXP3-SAE)

---

Create set of discretized policies $\mathbf{e} \in \mathcal{E} = [(1/\varepsilon)^d]$, where $\varepsilon = (d\sigma \log(T)/T)^{1/(d+2)}$.

Set parameters $\eta = \sqrt{\frac{\log(|\mathcal{E}|)}{T\lambda^2|\mathcal{E}|}}$, $\gamma = 2\eta\lambda|\mathcal{E}|$, and $\lambda = \sigma\sqrt{2\log T}$.

Initialize probability distribution $p_t(\mathbf{e}) = 1/|\mathcal{E}|, \forall \mathbf{e} \in \mathcal{E}$.

**for** $t \in [T]$ **do**

    Choose policy $\mathbf{e}_t$ from probability distribution $q_t(\mathbf{e}) = (1 - \gamma) \cdot p_t(\mathbf{e}) + \frac{\gamma}{|\mathcal{E}|}$.

    Observe $\mathbf{x}'_t$.

    Play action $a_{t,\mathbf{e}_t} = 1$ if $\langle \mathbf{e}_t, \mathbf{x}'_t \rangle \geq \delta\|\mathbf{e}_t\|_2$. Otherwise play action $a_{t,\mathbf{e}_t} = 0$.

    Observe reward $r_t(a_{t,\mathbf{e}_t})$.

    Update loss estimator for each policy $\mathbf{e} \in \mathcal{E}$: $\widehat{\ell}_t(\mathbf{e}) = (1 + \lambda - r_t(a_{t,\mathbf{e}_t})) \cdot \mathbb{1}\{\mathbf{e} = \mathbf{e}_t\}/q_t(\mathbf{e})$.

    Update probability distribution $\forall \mathbf{e} \in \mathcal{E}$: $p_{t+1}(\mathbf{e}) \propto p_t(\mathbf{e}) \cdot \exp\left(-\eta\widehat{\ell}_t(\mathbf{e})\right)$.

---

**Computational complexity** While both Algorithm 1 and Algorithm 3 have $\mathcal{O}(d^3)$ per-iteration computational complexity, Algorithm 4 must maintain and update a probability distribution over a grid of size exponential in $d$ at every time-step, making it hard to use in practice if $d$ is large. We view the design of computationally efficient algorithms for adversarially-chosen contexts as an important direction for future research.

**Extension to bandit feedback** Algorithm 4 may be extended to the bandit feedback setting by maintaining a grid over estimates of $\boldsymbol{\theta}^{(1)} - \boldsymbol{\theta}^{(0)}$ (instead of over $\boldsymbol{\theta}^{(1)}$). No further changes are required.

## 5 Conclusion

We study the problem of classification under incentives with apple tasting feedback. Such one-sided feedback is often what is observed in real-world strategic settings including lending and hiring. Our main result is a "greedy" algorithm (Algorithm 1) which achieves $\widetilde{\mathcal{O}}(\sqrt{T})$ strategic regret when the initial agent contexts are generated *stochastically*. The regret of Algorithm 1 depends on a constant $c_1(d, \delta)$ which scales exponentially in the context dimension, which may be problematic in settings for which the number of agents is small or unknown. To address this, we provide an algorithm (Algorithm 3) which combines Algorithm 1 with a strategy-aware version of the explore-then-commit algorithm using a doubling trick to achieve $\widetilde{\mathcal{O}}(\min\{\frac{\sqrt{dT}}{c_1(d,\delta)}, d \cdot T^{2/3}\})$ expected strategic regret whenever $T$ is unknown. Finally, we relax the assumption of stochastic contexts and allow for contexts to be generated adversarially. Algorithm 4 achieves $\widetilde{\mathcal{O}}(T^{\frac{d+1}{d+2}})$ expected strategic regret whenever agent contexts are generated adversarially by running EXP3 over a discretized grid of strategy-aware policies, but has exponential-in-$d$ per-round computational complexity. All of our results also apply to the more general setting of bandit feedback, under slight modifications to the algorithms. There are several directions for future work:

**Unclean data** The regret of Algorithm 1 depends on a constant which is exponentially large in $d$, due to the fact that it only learns using clean data (Condition 3.2). While learning using unclean data will generally produce an inconsistent estimator, it would be interesting to see if the principal could leverage this data to remove the dependence on this constant. Alternatively, lower bounds which show that using unclean data will not improve regret would also be interesting.

**Efficient algorithms for adversarial contexts** Our algorithm for adversarially-chosen agent contexts suffers exponential-in-$d$ per-round computational complexity, which makes it unsuitable for use in settings with high-dimensional contexts. Deriving polynomial-time algorithms with sublinear strategic regret for this setting is an exciting (but challenging) direction for future research.

**More than two actions** Finally, it would be interesting to extend our algorithms for strategic learning under bandit feedback to the setting in which the principal has *three or more* actions at their disposal. While prior work [29] implies an impossibility result for strategic regret minimization with three or more actions, other (relaxed) notions of optimality (e.g., sublinear *Stackelberg* regret; recall Definition 2.2) may still be possible.

## Acknowledgements

KH is supported in part by an NDSEG Fellowship. KH and ZSW are supported in part by the NSF FAI Award #1939606. For part of this work, CP was supported by a FODSI postdoctoral fellowship from UC Berkeley. The authors would like to thank the anonymous NeurIPS reviewers for valuable feedback.

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

# A  Useful concentration inequalities

**Theorem A.1** (Matrix Azuma, Tropp [54])**.** *Consider a self-adjoint matrix martingale $\{Y_s : s = 1, \ldots, t\}$ in dimension $d$, and let $\{X_s\}_{s \in [t]}$ be the associated difference sequence satisfying $\mathbb{E}_{s-1} X_s = 0_{d \times d}$ and $X_s^2 \preceq A_s^2$ for some fixed sequence $\{A_s\}_{s \in [t]}$ of self-adjoint matrices. Then for all $\alpha > 0$,*

$$\mathbb{P}\left(\lambda_{max}(Y_t - \mathbb{E}Y_t) \geq \alpha\right) \leq d \cdot \exp(-\alpha^2 / 8\sigma^2),$$

*where $\sigma^2 := \left\|\sum_{s=1}^{t} A_s^2\right\|_2$.*

**Theorem A.2** (A variant of Azuma's inequality for martingales with subgaussian tails, Shamir [51])**.** *Let $Z_1, Z_2, \ldots Z_t$ be a martingale difference sequence with respect to a sequence $W_1, W_2, \ldots, W_t$, and suppose there are constants $b > 1$, $c > 0$ such that for any $s$ and any $\alpha > 0$, it holds that*

$$\max\{\mathbb{P}(Z_s > \alpha | X_1, \ldots, X_{s-1}), \mathbb{P}(Z_s < -\alpha | X_1, \ldots, X_{s-1})\} \leq b \cdot \exp(-c\alpha^2).$$

*Then for any $\gamma > 0$, it holds with probability $1 - \gamma$ that*

$$\sum_{s=1}^{t} Z_s \leq \sqrt{\frac{28b \log(1/\gamma)}{cT}}.$$

# B  Proofs for Section 3

## B.1  Proof of Theorem 3.3

**Theorem B.1.** *Let $f_{U^d} : \mathcal{X} \to \mathbb{R}_{\geq 0}$ denote the density function of the uniform distribution over the $d$-dimensional unit sphere. If agent contexts are drawn from a distribution over the $d$-dimensional unit sphere with density function $f : \mathcal{X} \to \mathbb{R}_{\geq 0}$ such that $\frac{f(\mathbf{x})}{f_{U^d}(\mathbf{x})} \geq c_0 > 0, \forall \mathbf{x} \in \mathcal{X}$, then Algorithm 1 achieves the following performance guarantee:*

$$\text{Reg}(T) \leq 4d + \frac{8}{c_0 \cdot c_1(\delta, d) \cdot c_2(\delta, d)} \sqrt{14d\sigma^2 T \log(4dT/\gamma)}$$

*with probability $1 - \gamma$, where $0 < c_1(\delta, d) := \mathbb{P}_{\mathbf{x} \sim U^d}(\mathbf{x}[1] \geq \delta)$ and $0 < c_2(\delta, d) := \mathbb{E}_{\mathbf{x} \sim U^d}[\mathbf{x}[2]^2 | \mathbf{x}[1] \geq \delta]$.*

*Proof.* We start from the definition of strategic regret. Note that under apple tasting feedback, $\boldsymbol{\theta}^{(0)} = \mathbf{0}$.

$$
\begin{aligned}
\text{Reg}(T) &:= \sum_{t=1}^{T} \left\langle \boldsymbol{\theta}^{(a_t^*)} - \boldsymbol{\theta}^{(a_t)}, \mathbf{x}_t \right\rangle \\
&= \sum_{t=1}^{T} \left\langle \hat{\boldsymbol{\theta}}_t^{(a_t^*)} - \hat{\boldsymbol{\theta}}_t^{(a_t)}, \mathbf{x}_t \right\rangle + \left\langle \boldsymbol{\theta}^{(a_t^*)} - \hat{\boldsymbol{\theta}}_t^{(a_t^*)}, \mathbf{x}_t \right\rangle + \left\langle \hat{\boldsymbol{\theta}}_t^{(a_t)} - \boldsymbol{\theta}^{(a_t)}, \mathbf{x}_t \right\rangle \\
&\leq \sum_{t=1}^{T} \left| \left\langle \boldsymbol{\theta}^{(1)} - \hat{\boldsymbol{\theta}}_t^{(1)}, \mathbf{x}_t \right\rangle \right| + \left| \left\langle \boldsymbol{\theta}^{(0)} - \hat{\boldsymbol{\theta}}_t^{(0)}, \mathbf{x}_t \right\rangle \right| \\
&\leq \sum_{t=1}^{T} \left\| \boldsymbol{\theta}^{(1)} - \hat{\boldsymbol{\theta}}_t^{(1)} \right\|_2 \|\mathbf{x}_t\|_2 + \left\| \boldsymbol{\theta}^{(0)} - \hat{\boldsymbol{\theta}}_t^{(0)} \right\|_2 \|\mathbf{x}_t\|_2 \\
&\leq 2d + \sum_{t=2d+1}^{T} \left\| \boldsymbol{\theta}^{(1)} - \hat{\boldsymbol{\theta}}_t^{(1)} \right\|_2 + \left\| \boldsymbol{\theta}^{(0)} - \hat{\boldsymbol{\theta}}_t^{(0)} \right\|_2
\end{aligned}
$$

where the first inequality follows from Lemma B.2, the second inequality follows from the Cauchy-Schwarz, and the third inequality follows from the fact that the instantaneous regret at each time-step is at most 2 and we use the first $d$ rounds to bootstrap our OLS. The result follows by Lemma B.3, a union bound, and the fact that $\sum_{d+1}^{T} \sqrt{\frac{1}{t}} \leq 2\sqrt{T}$. $\qquad\square$

**Lemma B.2.**

$$\langle \widehat{\boldsymbol{\theta}}_t^{(a_t^*)} - \widehat{\boldsymbol{\theta}}_t^{(a_t)}, \mathbf{x}_t \rangle \leq 0$$

*Proof.* If $a_t = a_t^*$, the condition is satisfied trivially. If $a_t \neq a_t^*$, then either (1) $\langle \widehat{\boldsymbol{\theta}}_t^{(1)} - \widehat{\boldsymbol{\theta}}_t^{(0)}, \mathbf{x}_t' \rangle - \delta \| \widehat{\boldsymbol{\theta}}_t^{(1)} - \widehat{\boldsymbol{\theta}}_t^{(0)} \|_2 \geq 0$ and $\langle \boldsymbol{\theta}^{(1)} - \boldsymbol{\theta}^{(0)} \rangle < 0$ or (2) $\langle \widehat{\boldsymbol{\theta}}_t^{(1)} - \widehat{\boldsymbol{\theta}}_t^{(0)}, \mathbf{x}_t' \rangle - \delta \| \widehat{\boldsymbol{\theta}}_t^{(1)} - \widehat{\boldsymbol{\theta}}_t^{(0)} \|_2 < 0$ and $\langle \boldsymbol{\theta}^{(1)} - \boldsymbol{\theta}^{(0)} \rangle \geq 0$.

**Case 1:** $\langle \widehat{\boldsymbol{\theta}}_t^{(1)} - \widehat{\boldsymbol{\theta}}_t^{(0)}, \mathbf{x}_t' \rangle - \delta \| \widehat{\boldsymbol{\theta}}_t^{(1)} - \widehat{\boldsymbol{\theta}}_t^{(0)} \|_2 \geq 0$ and $\langle \boldsymbol{\theta}^{(1)} - \boldsymbol{\theta}^{(0)} \rangle < 0$. ($a_t^* = 0$, $a_t = 1$)
By Definition 2.1, we can rewrite

$$\langle \widehat{\boldsymbol{\theta}}_t^{(1)} - \widehat{\boldsymbol{\theta}}_t^{(0)}, \mathbf{x}_t' \rangle - \delta \| \widehat{\boldsymbol{\theta}}_t^{(1)} - \widehat{\boldsymbol{\theta}}_t^{(0)} \|_2 \geq 0$$

as

$$\langle \widehat{\boldsymbol{\theta}}_t^{(1)} - \widehat{\boldsymbol{\theta}}_t^{(0)}, \mathbf{x}_t \rangle + (\delta' - \delta) \| \widehat{\boldsymbol{\theta}}_t^{(1)} - \widehat{\boldsymbol{\theta}}_t^{(0)} \|_2 \geq 0$$

for some $\delta' \leq \delta$. Since $(\delta' - \delta) \| \widehat{\boldsymbol{\theta}}_t^{(1)} - \widehat{\boldsymbol{\theta}}_t^{(0)} \|_2 \leq 0$, $\langle \widehat{\boldsymbol{\theta}}_t^{(1)} - \widehat{\boldsymbol{\theta}}_t^{(0)}, \mathbf{x}_t \rangle \geq 0$ must hold.

**Case 2:** $\langle \widehat{\boldsymbol{\theta}}_t^{(1)} - \widehat{\boldsymbol{\theta}}_t^{(0)}, \mathbf{x}_t' \rangle - \delta \| \widehat{\boldsymbol{\theta}}_t^{(1)} - \widehat{\boldsymbol{\theta}}_t^{(0)} \|_2 < 0$ and $\langle \boldsymbol{\theta}^{(1)} - \boldsymbol{\theta}^{(0)} \rangle \geq 0$. ($a_t^* = 1$, $a_t = 0$)
Since modification did not help agent $t$ receive action $a_t = 1$, we can conclude that $\langle \widehat{\boldsymbol{\theta}}_t^{(1)} - \widehat{\boldsymbol{\theta}}_t^{(0)}, \mathbf{x}_t \rangle < 0$. $\qquad \square$

**Lemma B.3.** *Let $f_{U^d} : \mathcal{X} \to \mathbb{R}_{\geq 0}$ denote the density function of the uniform distribution over the $d$-dimensional unit sphere. If $T \geq d$ and agent contexts are drawn from a distribution over the $d$-dimensional unit sphere with density function $f : \mathcal{X} \to \mathbb{R}_{\geq 0}$ such that $\frac{f(\mathbf{x})}{f_{U^d}(\mathbf{x})} \geq c_0 \in \mathbb{R}_{>0}$, $\forall \mathbf{x} \in \mathcal{X}$, then the following guarantee holds under apple tasting feedback.*

$$\| \boldsymbol{\theta}^{(1)} - \hat{\boldsymbol{\theta}}_{t+1}^{(1)} \|_2 \leq \frac{2}{c_0 \cdot c_1(\delta, d) \cdot c_2(\delta, d)} \sqrt{\frac{14 d \sigma^2 \log(2d/\gamma_t)}{t}}$$

*with probability $1 - \gamma_t$.*

*Proof.* Let $\mathcal{I}_s^{(1)} = \{ \langle \hat{\boldsymbol{\theta}}_s^{(1)}, \mathbf{x}_s \rangle \geq \delta \| \hat{\boldsymbol{\theta}}_s^{(1)} \|_2 + r_0 \}$. Then, from the definition of $\boldsymbol{\theta}_{t+1}^{(1)}$ we have:

$$
\begin{aligned}
\hat{\boldsymbol{\theta}}_{t+1}^{(1)} &:= \left( \sum_{s=1}^t \mathbf{x}_s \mathbf{x}_s^\top \mathbb{1}\{\mathcal{I}_s^{(1)}\} \right)^{-1} \sum_{s=1}^t \mathbf{x}_s r_s(1) \mathbb{1}\{\mathcal{I}_s^{(1)}\} && \text{(closed form solution of OLS)} \\
&= \left( \sum_{s=1}^t \mathbf{x}_s \mathbf{x}_s^\top \mathbb{1}\{\mathcal{I}_s^{(1)}\} \right)^{-1} \sum_{s=1}^t \mathbf{x}_s (\mathbf{x}_s^\top \boldsymbol{\theta}^{(1)} + \epsilon_s) \mathbb{1}\{\mathcal{I}_s^{(1)}\} && \text{(plug in } r_s(1)) \\
&= \boldsymbol{\theta}^{(1)} + \left( \sum_{s=1}^t \mathbf{x}_s \mathbf{x}_s^\top \mathbb{1}\{\mathcal{I}_s^{(1)}\} \right)^{-1} \sum_{s=1}^t \mathbf{x}_s \epsilon_s \mathbb{1}\{\mathcal{I}_s^{(1)}\}
\end{aligned}
$$

Re-arranging the above and taking the $\ell_2$ norm on both sides we get:

$$
\begin{aligned}
\left\| \boldsymbol{\theta}^{(1)} - \hat{\boldsymbol{\theta}}_{t+1}^{(1)} \right\|_2 &= \left\| \left( \sum_{s=1}^t \mathbf{x}_s \mathbf{x}_s^\top \mathbb{1}\{\mathcal{I}_s^{(1)}\} \right)^{-1} \sum_{s=1}^t \mathbf{x}_s \epsilon_s \mathbb{1}\{\mathcal{I}_s^{(1)}\} \right\|_2 \\
&\leq \left\| \left( \sum_{s=1}^t \mathbf{x}_s \mathbf{x}_s^\top \mathbb{1}\{\mathcal{I}_s^{(1)}\} \right)^{-1} \right\|_2 \left\| \sum_{s=1}^t \mathbf{x}_s \epsilon_s \mathbb{1}\{\mathcal{I}_s^{(1)}\} \right\|_2 && \text{(Cauchy-Schwarz)} \\
&= \frac{\left\| \sum_{s=1}^t \mathbf{x}_s \epsilon_s \mathbb{1}\{\mathcal{I}_s^{(1)}\} \right\|_2}{\sigma_{min} \left( \sum_{s=1}^t \mathbf{x}_s \mathbf{x}_s^\top \mathbb{1}\{\mathcal{I}_s^{(1)}\} \right)} \\
&= \frac{\left\| \sum_{s=1}^t \mathbf{x}_s \epsilon_s \mathbb{1}\{\mathcal{I}_s^{(1)}\} \right\|_2}{\lambda_{min} \left( \sum_{s=1}^t \mathbf{x}_s \mathbf{x}_s^\top \mathbb{1}\{\mathcal{I}_s^{(1)}\} \right)}
\end{aligned}
$$

where for a matrix $M$, $\sigma_{\min}$ is the smallest singular value $\sigma_{\min}(M) := \min_{\|x\|=1} \|Mx\|$ and $\lambda_{\min}$ is the smallest eigenvalue. Note that the two are equal since the matrix $\sum_{s=1}^{t} \mathbf{x}_s \mathbf{x}_s^\top \mathbb{1}\{\mathcal{I}_s^{(1)}\}$ is PSD as the sum of PSD matrices (outer products induce PSD matrices). The final bound is obtained by applying Lemma B.4, Lemma B.5, and a union bound. $\qquad\square$

**Lemma B.4.** *The following bound holds on the $\ell_2$-norm of $\sum_{s=1}^{t} \mathbf{x}_s \epsilon_s \mathbb{1}\{\mathcal{I}_s^{(1)}\}$ with probability $1 - \gamma_t$:*

$$\left\| \sum_{s=1}^{t} \mathbf{x}_s \epsilon_s \mathbb{1}\{\mathcal{I}_s^{(1)}\} \right\|_2 \leq 2\sqrt{14 d \sigma^2 t \log(d/\gamma_t)}$$

*Proof.* Let $\mathbf{x}[i]$ denote the $i$-th coordinate of a vector $\mathbf{x}$. Observe that $\sum_{s=1}^{t} \epsilon_s \mathbf{x}_s[i] \mathbb{1}\{\mathcal{I}_s^{(1)}\}$ is a sum of martingale differences with $Z_s := \epsilon_s \mathbf{x}_s[i] \mathbb{1}\{\mathcal{I}_s^{(1)}\}$, $X_s := \sum_{s'=1}^{s} \epsilon_{s'} \mathbf{x}_{s'}[i] \mathbb{1}\{\mathcal{I}_{s'}^{(1)}\}$, and

$$\max\{\mathbb{P}(Z_s > \alpha | X_1, \ldots, X_{s-1}), \mathbb{P}(Z_s < -\alpha | X_1, \ldots, X_{s-1})\} \leq \exp(-\alpha^2/2\sigma^2).$$

By Theorem A.2,

$$\sum_{s=1}^{t} \epsilon_s \mathbf{x}_s[i] \mathbb{1}\{\mathcal{I}_s^{(1)}\} \leq 2\sqrt{14 \sigma^2 t \log(1/\gamma_i)}$$

with probability $1 - \gamma_i$. The desired result follows via a union bound and algebraic manipulation. $\qquad\square$

**Lemma B.5.** *The following bound holds on the minimum eigenvalue of $\sum_{s=1}^{t} \mathbf{x}_s \mathbf{x}_s^\top \mathbb{1}\{\mathcal{I}_s^{(1)}\}$ with probability $1 - \gamma_t$:*

$$\lambda_{min}\left( \sum_{s=1}^{t} \mathbf{x}_s \mathbf{x}_s^\top \mathbb{1}\{\mathcal{I}_s^{(1)}\} \right) \geq \frac{t}{c_0 \cdot c_1(\delta, d) \cdot c_2(\delta, d)} + 4\sqrt{2t \log(d/\gamma_t)}$$

*Proof.*

$$\lambda_{min}\left( \sum_{s=1}^{t} \mathbf{x}_s \mathbf{x}_s^\top \mathbb{1}\{\mathcal{I}_s^{(1)}\} \right)$$

$$\geq \lambda_{min}\left( \sum_{s=1}^{t} \mathbf{x}_s \mathbf{x}_s^\top \mathbb{1}\{\mathcal{I}_s^{(1)}\} - \mathbb{E}_{s-1}[\mathbf{x}_s \mathbf{x}_s^\top \mathbb{1}\{\mathcal{I}_s^{(1)}\}] \right) + \lambda_{min}\left( \sum_{s=1}^{t} \mathbb{E}_{s-1}[\mathbf{x}_s \mathbf{x}_s^\top \mathbb{1}\{\mathcal{I}_s^{(1)}\}] \right)$$

$$\geq \lambda_{min}\left( \sum_{s=1}^{t} \mathbf{x}_s \mathbf{x}_s^\top \mathbb{1}\{\mathcal{I}_s^{(1)}\} - \mathbb{E}_{s-1}[\mathbf{x}_s \mathbf{x}_s^\top \mathbb{1}\{\mathcal{I}_s^{(1)}\}] \right) + \sum_{s=1}^{t} \lambda_{min}\left( \mathbb{E}_{s-1}[\mathbf{x}_s \mathbf{x}_s^\top \mathbb{1}\{\mathcal{I}_s^{(1)}\}] \right) \quad (2)$$

where the inequalities follow from the fact that $\lambda_{min}(A + B) \geq \lambda_{min}(A) + \lambda_{min}(B)$ for two Hermitian matrices $A$, $B$. Note that the outer products form Hermitian matrices. Let $Y_t := \sum_{s=1}^{t} \mathbf{x}_s \mathbf{x}_s^\top \mathbb{1}\{\mathcal{I}_s^{(1)}\} - \mathbb{E}_{s-1}[\mathbf{x}_s \mathbf{x}_s^\top \mathbb{1}\{\mathcal{I}_s^{(1)}\}]$. Note that by the tower rule, $\mathbb{E}Y_t = \mathbb{E}Y_0 = 0$. Let $-X_s := \mathbb{E}_{s-1}[\mathbf{x}_s \mathbf{x}_s^\top \mathbb{1}\{\mathcal{I}_s^{(1)}\}]$, then $\mathbb{E}_{s-1}[-X_s] = 0$, and $(-X_s)^2 \preceq 4I_d$. By Theorem A.1,

$$\mathbb{P}(\lambda_{max}(-Y_t) \geq \alpha) \leq d \cdot \exp(-\alpha^2/32t).$$

Since $-\lambda_{max}(-Y_t) = \lambda_{min}(Y_t)$,

$$\mathbb{P}(\lambda_{max}(Y_t) \leq \alpha) \leq d \cdot \exp(-\alpha^2/32t).$$

Therefore, $\lambda_{min}(Y_t) \geq 4\sqrt{2t \log(d/\gamma_t)}$ with probability $1 - \gamma_t$. We now turn our attention to lower bounding $\lambda_{min}(\mathbb{E}_{s-1}[\mathbf{x}_s \mathbf{x}_s^\top \mathbb{1}\{\mathcal{I}_s^{(1)}\}])$.

$$\lambda_{min}(\mathbb{E}_{s-1}[\mathbf{x}_s\mathbf{x}_s^\top \mathbb{1}\{\mathcal{I}_s^{(1)}\}]) := \min_{\boldsymbol{\omega}\in S^{d-1}} \boldsymbol{\omega}^\top \mathbb{E}_{s-1}[\mathbf{x}_s\mathbf{x}_s^\top \mathbb{1}\{\mathcal{I}_s^{(1)}\}]\boldsymbol{\omega}$$

$$= \min_{\boldsymbol{\omega}\in S^{d-1}} \boldsymbol{\omega}^\top \left(\int \mathbf{x}_s\mathbf{x}_s^\top \mathbb{1}\{\mathcal{I}_s^{(1)}\}f(\mathbf{x}_s)d\mathbf{x}_s\right)\boldsymbol{\omega}$$

$$= \min_{\boldsymbol{\omega}\in S^{d-1}} \boldsymbol{\omega}^\top \left(\int \mathbf{x}_s\mathbf{x}_s^\top \mathbb{1}\{\mathcal{I}_s^{(1)}\}f(\mathbf{x}_s)\cdot \frac{f_{U^d}(\mathbf{x}_s)}{f(\mathbf{x}_s)}d\mathbf{x}_s\right)\boldsymbol{\omega}$$

$$\geq c_0\cdot \min_{\boldsymbol{\omega}\in S^{d-1}} \boldsymbol{\omega}^\top \mathbb{E}_{s-1,U^d}[\mathbf{x}_s\mathbf{x}_s^\top \mathbb{1}\{\mathcal{I}_s^{(1)}\}]\boldsymbol{\omega}$$

$$= c_0 \min_{\boldsymbol{\omega}\in S^{d-1}} \mathbb{E}_{s-1,U^d}[\langle\boldsymbol{\omega},\mathbf{x}_s\rangle^2|\langle\widehat{\boldsymbol{\beta}}_s,\mathbf{x}_s\rangle \geq \delta\|\widehat{\boldsymbol{\beta}}_s\|_2]\cdot \underbrace{\mathbb{P}_{s-1,U^d}(\langle\widehat{\boldsymbol{\beta}}_s,\mathbf{x}_s\rangle \geq \delta\|\widehat{\boldsymbol{\beta}}_s\|_2)}_{c_1(\delta,d)}$$

$$= c_0\cdot c_1(\delta,d) \min_{\boldsymbol{\omega}\in S^{d-1}} \mathbb{E}_{s-1,U^d}[\langle\boldsymbol{\omega},\mathbf{x}_s\rangle^2|\langle\widehat{\boldsymbol{\beta}}_s,\mathbf{x}_s\rangle \geq \delta\|\widehat{\boldsymbol{\beta}}_s\|_2] \qquad (3)$$

Throughout the remainder of the proof, we surpress the dependence on $U^d$ and note that unless stated otherwise, all expectations are taken with respect to $U^d$. Let $B_s \in \mathbb{R}^{d\times d}$ be the orthonormal matrix such that the first column is $\widehat{\boldsymbol{\beta}}_s/\|\widehat{\boldsymbol{\beta}}_s\|_2$. Note that $B_s e_1 = \widehat{\boldsymbol{\beta}}_s/\|\widehat{\boldsymbol{\beta}}_s\|_2$ and $B_s\mathbf{x}\sim U^d$.

$$\mathbb{E}_{s-1}[\mathbf{x}_s\mathbf{x}_s^\top|\langle\widehat{\boldsymbol{\beta}}_s,\mathbf{x}_s\rangle \geq \delta\|\widehat{\boldsymbol{\beta}}_s\|] = \mathbb{E}_{s-1}[(B_s\mathbf{x}_s)(B_s\mathbf{x}_s)^\top|\langle\widehat{\boldsymbol{\beta}}_s,B_s\mathbf{x}_s\rangle \geq \delta]$$

$$= B_s\mathbb{E}_{s-1}[\mathbf{x}_s\mathbf{x}_s^\top|\mathbf{x}_s^\top B_s^\top\|\widehat{\boldsymbol{\beta}}_s\|_2 B_s e_1 \geq \delta\cdot\|\widehat{\boldsymbol{\beta}}_s\|_2]B_s^\top$$

$$= B_s\mathbb{E}_{s-1}[\mathbf{x}_s\mathbf{x}_s^\top|\mathbf{x}_s[1]\geq \delta]B_s^\top$$

Observe that for $j\neq 1$, $i\neq j$, $\mathbb{E}[\mathbf{x}_s[j]\mathbf{x}_s[i]|\mathbf{x}_s[1]\geq\delta] = 0$. Therefore,

$$\mathbb{E}_{s-1}[\mathbf{x}_s\mathbf{x}_s^\top|\langle\widehat{\boldsymbol{\beta}}_s,\mathbf{x}_s\rangle \geq \delta\|\widehat{\boldsymbol{\beta}}_s] = B_s(\mathbb{E}[\mathbf{x}_s[2]^2|\mathbf{x}_s[1]\geq\delta]I_d$$

$$+ (\mathbb{E}[\mathbf{x}_s[1]^2|\mathbf{x}_s[1]\geq\delta] - \mathbb{E}[\mathbf{x}_s[2]^2|\mathbf{x}_s[1]\geq\delta])\mathbf{e}_1\mathbf{e}_1^\top)B_s^\top$$

$$= \mathbb{E}[\mathbf{x}_s[2]^2|\mathbf{x}_s[1]\geq\delta]I_d$$

$$+ (\mathbb{E}[\mathbf{x}_s[1]^2|\mathbf{x}_s[1]\geq\delta] - \mathbb{E}[\mathbf{x}_s[2]^2|\mathbf{x}_s[1]\geq\delta])\frac{\widehat{\boldsymbol{\beta}}_s}{\|\widehat{\boldsymbol{\beta}}_s\|_2}\left(\frac{\widehat{\boldsymbol{\beta}}_s}{\|\widehat{\boldsymbol{\beta}}_s\|_2}\right)^\top$$

and

$$\lambda_{min}(\mathbb{E}_{s-1}[\mathbf{x}_s\mathbf{x}_s^\top\mathbb{1}\{\mathcal{I}_s^{(1)}\}]) \geq c_0\cdot c_1(\delta,d) \min_{\boldsymbol{\omega}\in S^{d-1}} \left(\mathbb{E}[\mathbf{x}_s[2]^2|\mathbf{x}_s[1]\geq\delta]\|\boldsymbol{\omega}\|_2\right.$$

$$+ (\mathbb{E}[\mathbf{x}_s[1]^2|\mathbf{x}_s[1]\geq\delta] - \mathbb{E}[\mathbf{x}_s[2]^2|\mathbf{x}_s[1]\geq\delta])\left\langle\boldsymbol{\omega},\frac{\widehat{\boldsymbol{\beta}}_s}{\|\widehat{\boldsymbol{\beta}}_s\|_2}\right\rangle^2)$$

$$\geq c_0\cdot c_1(\delta,d)\cdot c_2(\delta,d)$$

$\square$

**Lemma B.6.** *For sufficiently large values of d,*

$$c_1(\delta,d) \geq \Theta\left(\frac{(1-\delta)^{d/2}}{d}\right).$$

*Proof.* Lemma B.6 is obtained via a similar argument to Theorem 2.7 in Blum et al. [15]. As in Blum et al. [15], we are interested in the volume of a hyperspherical cap. However, we are interested in a lower-bound, not an upper-bound (as is the case in [15]). Let A denote the portion of the $d$-dimensional hypersphere with $\mathbf{x}[1]\geq\frac{\sqrt{c}}{d-1}$ and let H denote the upper hemisphere.

$$c_1(\delta,d) := \mathbb{P}_{\mathbf{x}\sim U^d}(\mathbf{x}[1]\geq\delta) = \frac{vol(A)}{vol(H)}$$

In order to lower-bound $c_1(\delta,d)$, it suffices to lower bound $vol(A)$ and upper-bound $vol(H)$. In what follows, let $V(d)$ denote the volume of the $d$-dimensional hypersphere with radius 1.

**Lower-bounding** $vol(A)$**:** As in [15], to calculate the volume of A, we integrate an incremental volume that is a disk of width $d\mathbf{x}[1]$ and whose face is a ball of dimension $d-1$ and radius $\sqrt{1-\mathbf{x}[1]^2}$. The surface area of the disk is $(1-\mathbf{x}[1]^2)^{\frac{d-1}{2}}V(d-1)$ and the volume above the slice $\mathbf{x}[1] \geq \delta$ is

$$vol(A) = \int_\delta^1 (1-\mathbf{x}[1]^2)^{\frac{d-1}{2}} V(d-1) d\mathbf{x}[1]$$

To get a lower bound on the integral, we use the fact that $1 - x^2 \geq 1 - x$ for $x \in [0,1]$. The integral now takes the form

$$vol(A) \geq \int_\delta^1 (1-\mathbf{x}[1])^{\frac{d-1}{2}} V(d-1) d\mathbf{x}[1] = \frac{V(d-1)}{d+1} \cdot 2(1-\delta)^{\frac{d+1}{2}}$$

**Upper-bounding** $vol(H)$**:** We can obtain an exact expression for $vol(H)$ in terms of $V(d-1)$ using the recursive relationship between $V(d)$ and $V(d-1)$:

$$vol(H) = \frac{1}{2}V(d) = \frac{\sqrt{\pi}}{2} \frac{\Gamma(\frac{d}{2} + \frac{1}{2})}{\Gamma(\frac{d}{2}+1)} V(d-1)$$

Plugging in our bounds for $vol(A)$, $vol(H)$ and simplifying, we see that

$$c_1(\delta, d) \geq \frac{(1-\delta)^{\frac{d+1}{2}}}{\sqrt{\pi}(d+1)} \frac{\Gamma(\frac{d}{2}+1)}{\Gamma(\frac{d}{2}+\frac{1}{2})} = \Theta\left(\frac{(1-\delta)^{\frac{d+1}{2}}}{d+1}\right)$$

where the equality follows from Stirling's approximation. $\qquad\square$

**Lemma B.7.** *The following bound holds on $c_2(\delta, d)$:*

$$c_2(\delta, d) \geq \frac{1}{3d}\left(\frac{3}{4} - \frac{1}{2}\delta - \frac{1}{4}\delta^2\right)^3.$$

*Proof.* We begin by computing $\mathbb{E}[\mathbf{x}[2]^2 | \mathbf{x}[1] = \delta']$, for $\delta' \in (0,1)$. If $\mathbf{x}[1] = \delta'$, then $\mathbf{x}[2]^2 + \ldots + \mathbf{x}[d]^2 \leq 1 - (\delta')^2$. Using this fact, we see that

$$\mathbb{E}[\mathbf{x}[2]^2 | \mathbf{x}[1] = \delta'] = \frac{1}{d}\mathbb{E}_{r \sim Unif[0, 1-(\delta')^2]}[r^2] = \frac{1}{3d}(1-(\delta')^2)^3.$$

Since $\mathbb{E}[\mathbf{x}[2]^2 | \mathbf{x}[1] \geq \delta] \geq \mathbb{E}[\mathbf{x}[2]^2 | \mathbf{x}[1] = \delta + \frac{1-\delta}{2}]$,

$$\mathbb{E}[\mathbf{x}[2]^2 | \mathbf{x}[1] \geq \delta] \geq \mathbb{E}\left[\mathbf{x}[2]^2 | \mathbf{x}[1] = \frac{\delta+1}{2}\right] = \frac{1}{3d}\left(\frac{3}{4} - \frac{1}{2}\delta - \frac{1}{4}\delta^2\right)^3$$

$\qquad\square$

## B.2   Proof of Proposition 3.4

**Proposition B.8.** *For any sequence of linear threshold policies $\boldsymbol{\beta}_1, \ldots, \boldsymbol{\beta}_T$,*

$$\mathbb{E}_{\mathbf{x}_1,\ldots,\mathbf{x}_T \sim U^d}\left[\sum_{t=1}^T \mathbb{1}\{\langle \mathbf{x}_t, \boldsymbol{\beta}_t\rangle \geq \delta\|\boldsymbol{\beta}_t\|_2\}\right] = T \cdot \mathbb{P}_{\mathbf{x} \sim U^d}(\mathbf{x}[1] \geq \delta)$$

*Proof.* Let $B_t \in \mathbb{R}^{d \times d}$ be the orthonormal matrix such that the first column is $\boldsymbol{\beta}_t / \|\boldsymbol{\beta}_t\|_2$. Note that $B_t \mathbf{x} \sim U^d$ if $\mathbf{x} \sim U^d$ and $B_t e_1 = \boldsymbol{\beta}_t / \|\boldsymbol{\beta}_t\|_2$.

$$
\begin{aligned}
\mathbb{E}_{\mathbf{x}_1,\dots,\mathbf{x}_T \sim U^d}\Big[\sum_{t=1}^{T} \mathbb{1}\{\langle \mathbf{x}_t, \boldsymbol{\beta}_t \rangle \geq \delta \|\boldsymbol{\beta}_t\|_2\}\Big] &= \sum_{t=1}^{T} \mathbb{P}_{\mathbf{x}_t \sim U^d}(\langle \mathbf{x}_t, \boldsymbol{\beta}_t \rangle \geq \delta \|\boldsymbol{\beta}_t\|_2) \\
&= \sum_{t=1}^{T} \mathbb{P}_{\mathbf{x}_t \sim U^d}(\langle B_t \mathbf{x}_t, \boldsymbol{\beta}_t \rangle \geq \delta \|\boldsymbol{\beta}_t\|_2) \\
&= \sum_{t=1}^{T} \mathbb{P}_{\mathbf{x}_t \sim U^d}(\mathbf{x}_t^\top B_t^\top \|\boldsymbol{\beta}_t\|_2 B_t e_1 \geq \delta \|\boldsymbol{\beta}_t\|_2) \\
&= \sum_{t=1}^{T} \mathbb{P}_{\mathbf{x}_t \sim U^d}(\mathbf{x}_t^\top I_d e_1 \geq \delta \|\|_2) \\
&= T \cdot \mathbb{P}_{\mathbf{x} \sim U^d}(\mathbf{x}[1] \geq \delta \|\|_2)
\end{aligned}
$$

$\square$

## B.3 Explore-Then-Commit Analysis

**Theorem B.9.** *Let $f_{U^d} : \mathcal{X} \to \mathbb{R}_{\geq 0}$ denote the density function of the uniform distribution over the $d$-dimensional unit sphere. If agent contexts are drawn from a distribution over the $d$-dimensional unit sphere with density function $f : \mathcal{X} \to \mathbb{R}_{\geq 0}$ such that $\frac{f(\mathbf{x})}{f_U(\mathbf{x})} \geq c_0 > 0, \forall \mathbf{x} \in \mathcal{X}$, then Algorithm 2 achieves the following performance guarantee*

$$
\mathrm{Reg}_{\mathrm{ETC}}(T) \leq \frac{8 \cdot 63^{1/3}}{c_0} d\sigma^{2/3} T^{2/3} \log^{1/3}(4d/\gamma)
$$

*with probability $1 - \gamma$ if $T_0 := 4 \cdot 63^{1/3} \sigma^{2/3} d T^{2/3} \log^{1/3}(4d/\gamma)$.*

*Proof.*

$$
\begin{aligned}
\mathrm{Reg}_{\mathrm{ETC}}(T) &:= \sum_{t=1}^{T} \langle \boldsymbol{\theta}^{(a_t^*)} - \boldsymbol{\theta}^{(a_t)}, \mathbf{x}_t \rangle \\
&\leq T_0 + \sum_{t=1}^{T} \langle \boldsymbol{\theta}^{(a_t^*)} - \boldsymbol{\theta}^{(a_t)}, \mathbf{x}_t \rangle \\
&= T_0 + \sum_{t=T_0+1}^{T} \langle \hat{\boldsymbol{\theta}}_{T_0/2}^{(a_t^*)} - \hat{\boldsymbol{\theta}}_{T_0/2}^{(a_t)}, \mathbf{x}_t \rangle + \langle \boldsymbol{\theta}^{(a_t^*)} - \hat{\boldsymbol{\theta}}_{T_0/2}^{(a_t^*)}, \mathbf{x}_t \rangle + \langle \hat{\boldsymbol{\theta}}_{T_0/2}^{(a_t)} - \boldsymbol{\theta}^{(a_t)}, \mathbf{x}_t \rangle \\
&\leq T_0 + \sum_{t=T_0+1}^{T} |\langle \boldsymbol{\theta}^{(1)} - \hat{\boldsymbol{\theta}}_{T_0/2}^{(1)}, \mathbf{x}_t \rangle| + |\langle \boldsymbol{\theta}^{(0)} - \hat{\boldsymbol{\theta}}_{T_0/2}^{(0)}, \mathbf{x}_t \rangle| \\
&\leq T_0 + \sum_{t=T_0+1}^{T} \|\boldsymbol{\theta}^{(1)} - \hat{\boldsymbol{\theta}}_{T_0/2}^{(1)}\|_2 \|\mathbf{x}_t\|_2 + \|\boldsymbol{\theta}^{(0)} - \hat{\boldsymbol{\theta}}_{T_0/2}^{(0)}\|_2 \|\mathbf{x}_t\|_2 \\
&\leq T_0 + T \cdot \|\boldsymbol{\theta}^{(1)} - \hat{\boldsymbol{\theta}}_{T_0/2}^{(1)}\|_2 + T \cdot \|\boldsymbol{\theta}^{(0)} - \hat{\boldsymbol{\theta}}_{T_0/2}^{(0)}\|_2 \\
&\leq T_0 + T \cdot \frac{24d}{c_0} \sqrt{\frac{7d\sigma^2 \log(4d/\gamma)}{T_0}}
\end{aligned}
$$

with probability $1 - \gamma$, where the last inequality follows from Lemma B.10 and a union bound. The result follows from picking $T_0 = 4 \cdot 63^{1/3} d\sigma^{2/3} T^{2/3} \log^{1/3}(4d/\gamma)$. $\square$

**Lemma B.10.** *Let $f_{U^d} : \mathcal{X} \to \mathbb{R}_{\geq 0}$ denote the density function of the uniform distribution over the $d$-dimensional unit sphere. If $T \geq 2d$ and agent contexts are drawn from a distribution over the*

*d-dimensional unit sphere with density function $f : \mathcal{X} \to \mathbb{R}_{\geq 0}$ such that $\frac{f(\mathbf{x})}{f_U(\mathbf{x})} \geq c_0 > 0$, $\forall \mathbf{x} \in \mathcal{X}$, then the following guarantee holds for $a \in \{0, 1\}$.*

$$\|\boldsymbol{\theta}^{(a)} - \hat{\boldsymbol{\theta}}^{(a)}_{t+1}\|_2 \leq \frac{12d}{c_0} \sqrt{\frac{7d\sigma^2 \log(2d/\gamma_t)}{T_0}}$$

*with probability $1 - \gamma_t$.*

*Proof.* Observe that

$$\hat{\boldsymbol{\theta}}^{(a)}_{T_0/2} := \left(\sum_{s=1}^{T_0/2} \mathbf{x}_{s+k}\mathbf{x}_{s+k}^\top\right)^{-1} \sum_{s=1}^{T_0/2} \mathbf{x}_{s+k} r^{(a)}_{s+k}$$

$$= \left(\sum_{s=1}^{T_0/2} \mathbf{x}_{s+k}\mathbf{x}_{s+k}^\top\right)^{-1} \sum_{s=1}^{T_0/2} \mathbf{x}_{s+k}(\mathbf{x}_{s+k}^\top \boldsymbol{\theta}^{(a)} + \epsilon_{s+k})$$

$$= \boldsymbol{\theta}^{(a)} + \left(\sum_{s=1}^{T_0/2} \mathbf{x}_{s+k}\mathbf{x}_{s+k}^\top\right)^{-1} \sum_{s=1}^{T_0/2} \mathbf{x}_{s+k}\epsilon_{s+k}$$

where $k = 0$ if $a = 0$ and $k = T_0$ if $a = 1$. Therefore,

$$\|\boldsymbol{\theta}^{(a)} - \hat{\boldsymbol{\theta}}^{(a)}_{T_0/2}\|_2 = \left\| \left(\sum_{s=1}^{T_0/2} \mathbf{x}_{s+k}\mathbf{x}_{s+k}^\top\right)^{-1} \sum_{s=1}^{T_0/2} \mathbf{x}_{s+k}\epsilon_{s+k} \right\|_2$$

$$\leq \left\| \left(\sum_{s=1}^{T_0/2} \mathbf{x}_{s+k}\mathbf{x}_{s+k}^\top\right)^{-1} \right\|_2 \left\| \sum_{s=1}^{T_0/2} \mathbf{x}_{s+k}\epsilon_{s+k} \right\|_2$$

$$= \frac{\left\| \sum_{s=1}^{T_0/2} \mathbf{x}_{s+k}\epsilon_{s+k} \right\|_2}{\sigma_{min}(\sum_{s=1}^{T_0/2} \mathbf{x}_{s+k}\mathbf{x}_{s+k}^\top)}$$

$$= \frac{\left\| \sum_{s=1}^{T_0/2} \mathbf{x}_{s+k}\epsilon_{s+k} \right\|_2}{\lambda_{min}(\sum_{s=1}^{T_0/2} \mathbf{x}_{s+k}\mathbf{x}_{s+k}^\top)}$$

The desired result is obtained by applying Lemma B.11, Lemma B.12, and a union bound. $\qquad\square$

**Lemma B.11.** *The following bound holds on the $\ell_2$-norm of $\sum_{s=1}^{T_0/2} \mathbf{x}_{s+k}\epsilon_{s+k}$ with probability $1 - \gamma$:*

$$\left\| \sum_{s=1}^{T_0/2} \mathbf{x}_{s+k}\epsilon_{s+k} \right\|_2 \leq 2\sqrt{7d\sigma^2 T_0 \log(d/\gamma)}$$

*Proof.* Observe that $\sum_{s=1}^{T_0/2} \epsilon_{k+s}\mathbf{x}_{k+s}[i]$ is a sum of martingale differences with $Z_{k+s} := \epsilon_{k+s}\mathbf{x}_{k+s}[i]$, $X_{k+s} := \sum_{s'=1}^{s} \epsilon_{k+s'}\mathbf{x}_{k+s'}[i]$, and

$$\max\{\mathbb{P}(Z_{k+s} > \alpha | X_{k+1}, \ldots, X_{k+s-1}), \mathbb{P}(Z_{k+s} < -\alpha | X_{k+1}, \ldots, X_{k+s-1})\} \leq \cdot \exp(-\alpha^2/2\sigma^2).$$

By Theorem A.2,

$$\sum_{s=1}^{T_0/2} \epsilon_{k+s}\mathbf{x}_{k+s}[i] \leq 2\sqrt{7\sigma^2 T_0 \log(1/\gamma_i)}$$

with probability $1 - \gamma_i$. The desired result follows via a union bound and algebraic manipulation. $\qquad\square$

**Lemma B.12.** *The following bound holds on the minimum eigenvalue of $\sum_{s=1}^{T_0/2} \mathbf{x}_{s+k}\mathbf{x}_{s+k}^\top$ with probability $1 - \gamma$:*

$$\lambda_{min}\left(\sum_{s=1}^{T_0/2} \mathbf{x}_{s+k}\mathbf{x}_{s+k}^\top\right) \geq \frac{T_0}{6d} + 4\sqrt{T_0 \log(d/\gamma)}$$

*Proof.*

$$\lambda_{min}(\sum_{s=1}^{T_0/2} \mathbf{x}_{s+k}\mathbf{x}_{s+k}^\top) \geq \lambda_{min}(\sum_{s=1}^{T_0/2} \mathbf{x}_{s+k}\mathbf{x}_{s+k}^\top - \mathbb{E}[\mathbf{x}_{s+k}\mathbf{x}_{s+k}^\top]) + \lambda_{min}(\sum_{s=1}^{T_0/2} \mathbb{E}[\mathbf{x}_{s+k}\mathbf{x}_{s+k}^\top])$$

$$\geq \lambda_{min}(\sum_{s=1}^{T_0/2} \mathbf{x}_{s+k}\mathbf{x}_{s+k}^\top - \mathbb{E}[\mathbf{x}_{s+k}\mathbf{x}_{s+k}^\top]) + \sum_{s=1}^{T_0/2} \lambda_{min}(\mathbb{E}[\mathbf{x}_{s+k}\mathbf{x}_{s+k}^\top])$$

where the inequalities follow from the fact that $\lambda_{min}(A + B) \geq \lambda_{min}(A) + \lambda_{min}(B)$ for two Hermitian matrices $A$, $B$. Let $Y_{T_0/2} := \sum_{s=1}^{T_0/2} \mathbf{x}_{s+k}\mathbf{x}_{s+k}^\top - \mathbb{E}[\mathbf{x}_{s+k}\mathbf{x}_{s+k}^\top]$. Note that $\mathbb{E}Y_{T_0/2} = \mathbb{E}Y_0 = 0$, $-X_{s+k} := \mathbb{E}[\mathbf{x}_{s+k}\mathbf{x}_{s+k}^\top]$, $\mathbb{E}[-X_{s+k}] = 0$, and $(-X_{s+k})^2 \preceq 4I_d$. By Theorem A.1,

$$\mathbb{P}(\lambda_{max}(-Y_{T_0/2}) \geq \alpha) \leq d \cdot \exp(-\alpha^2/16T_0).$$

Since $-\lambda_{max}(-Y_{T_0/2}) = \lambda_{min}(Y_{T_0/2})$,

$$\mathbb{P}(\lambda_{max}(Y_{T_0/2}) \leq \alpha) \leq d \cdot \exp(-\alpha^2/16T_0).$$

Therefore, $\lambda_{min}(Y_{T_0/2}) \geq 4\sqrt{T_0 \log(d/\gamma)}$ with probability $1 - \gamma$. We now turn our attention to lower bounding $\lambda_{min}(\mathbb{E}[\mathbf{x}_{s+k}\mathbf{x}_{s+k}^\top])$.

$$\lambda_{min}(\mathbb{E}[\mathbf{x}_{s+k}\mathbf{x}_{s+k}^\top]) := \min_{\boldsymbol{\omega} \in S^{d-1}} \boldsymbol{\omega}^\top \mathbb{E}[\mathbf{x}_{s+k}\mathbf{x}_{s+k}^\top]\boldsymbol{\omega}$$

$$= \min_{\boldsymbol{\omega} \in S^{d-1}} \boldsymbol{\omega}^\top \frac{1}{3d}I_d\boldsymbol{\omega}$$

$$= \frac{1}{3d}$$

$\square$

## B.4 Proof of Theorem 3.5

**Theorem B.13.** *Let* $\text{Reg}_{\text{OLS}}(T)$ *be the strategic regret of Algorithm 1 and* $\text{Reg}_{\text{ETC}}(T)$ *be the strategic regret of Algorithm 2. The expected strategic regret of Algorithm 3 is*

$$\mathbb{E}[\text{Reg}(T)] \leq 4 \cdot \min\{\mathbb{E}[\text{Reg}_{\text{OLS}}(T)], \mathbb{E}[\text{Reg}_{\text{ETC}}(T)]\}$$

*Proof.* **Case 1:** $T < \tau^*$ From Theorem B.9, we know that

$$\text{Reg}_{\text{ETC}}(\tau_i) \leq \frac{8 \cdot 63^{1/3}}{c_0} d\sigma^{2/3}\tau_i^{2/3} \log^{1/3}(4d\tau_i^2)$$

with probability $1 - 1/\tau_i^2$. Therefore,

$$\mathbb{E}[\text{Reg}_{\text{ETC}}(\tau_i)] \leq \frac{8 \cdot 63^{1/3}}{c_0} d\sigma^{2/3}\tau_i^{2/3} \log^{1/3}(4d\tau_i^2) + \frac{2}{\tau_i}$$

Observe that $\sum_{j=1}^{i-1} \mathbb{E}[\text{Reg}_{\text{ETC}}(\tau_j)] \leq \mathbb{E}[\text{Reg}_{\text{ETC}}(\tau_i)]$. Suppose $\tau_{i-1} \leq T \leq \tau_i$ for some $i$. Under such a scenario,

$$\mathbb{E}[\text{Reg}(T)] \leq 2\mathbb{E}[\text{Reg}_{\text{ETC}}(\tau_i)]$$
$$\leq 2\mathbb{E}[\text{Reg}_{\text{ETC}}(2T)]$$
$$\leq 4\mathbb{E}[\text{Reg}_{\text{ETC}}(T)]$$

**Case 2:** $T \geq \tau^*$ Let $t^*$ denote the actual switching time of Algorithm 3.

$$\text{Reg}(T) := \sum_{t=1}^{t^*} \langle \boldsymbol{\theta}^{(a_t^*)} - \boldsymbol{\theta}^{(a_t)}, \mathbf{x}_t \rangle + \sum_{t=t^*+1}^{T} \langle \boldsymbol{\theta}^{(a_t^*)} - \boldsymbol{\theta}^{(a_t)}, \mathbf{x}_t \rangle$$

$$\begin{aligned}
\mathbb{E}[\text{Reg}(T)] &\leq 2 \cdot \mathbb{E}[\text{Reg}_{\text{ETC}}(t^*)] + \mathbb{E}[\text{Reg}_{\text{OLS}}(T - t^*)] \\
&\leq 2 \cdot \mathbb{E}[\text{Reg}_{\text{OLS}}(t^*)] + \mathbb{E}[\text{Reg}_{\text{OLS}}(T)] \\
&\leq 2 \cdot \mathbb{E}[\text{Reg}_{\text{OLS}}(\tau^*)] + \mathbb{E}[\text{Reg}_{\text{OLS}}(T)] \\
&\leq 3 \cdot \mathbb{E}[\text{Reg}_{\text{OLS}}(T)]
\end{aligned}$$

where the first line follows from case 1, the second line follows from the fact that $t^* \leq \tau^*$ (and so $\mathbb{E}[\text{Reg}_{\text{ETC}}(t^*)] \leq \mathbb{E}[\text{Reg}_{\text{OLS}}(t^*)]$), the third line follows from the fact that $t^* \leq \tau^*$, and the fourth line follows from the fact that $T \geq \tau^*$. $\qquad\square$

### B.5 Inconsistency of OLS when using all data

**Theorem B.14.** $\lim_{t\to\infty} \hat{\boldsymbol{\theta}}_{t+1}^{(1)} = \boldsymbol{\theta}_*^{(1)}$ *if and only if* $\lim_{t\to\infty} \sum_{s=1}^{t} \mathbf{x}_s' \mathbf{x}_s'^\top \mathbb{1}\{a_s = 1\} = \lim_{t\to\infty} \sum_{s=1}^{t} \mathbf{x}_s' \mathbf{x}_s^\top \mathbb{1}\{a_s = 1\}$.

*Proof.*

$$\begin{aligned}
\lim_{t\to\infty} \hat{\boldsymbol{\theta}}_{t+1}^{(1)} &:= \lim_{t\to\infty} \left( \sum_{s=1}^{t} \mathbf{x}_s' \mathbf{x}_s'^\top \mathbb{1}\{a_s = 1\} \right)^{-1} \sum_{s=1}^{t} \mathbf{x}_s' r_s^{(1)} \mathbb{1}\{a_s = 1\} \\
&= \lim_{t\to\infty} \left( \sum_{s=1}^{t} \mathbf{x}_s' \mathbf{x}_s'^\top \mathbb{1}\{a_s = 1\} \right)^{-1} \sum_{s=1}^{t} \mathbf{x}_s' (\mathbf{x}_s^\top \boldsymbol{\theta}_*^{(1)} + \epsilon_s) \mathbb{1}\{a_s = 1\} \\
&= \lim_{t\to\infty} \left( \sum_{s=1}^{t} \mathbf{x}_s' \mathbf{x}_s'^\top \mathbb{1}\{a_s = 1\} \right)^{-1} \sum_{s=1}^{t} \mathbf{x}_s' \mathbf{x}_s^\top \boldsymbol{\theta}_*^{(1)} \mathbb{1}\{a_s = 1\} \\
&\quad + \lim_{t\to\infty} \left( \sum_{s=1}^{t} \mathbf{x}_s' \mathbf{x}_s'^\top \mathbb{1}\{a_s = 1\} \right)^{-1} \sum_{s=1}^{t} \mathbf{x}_s' \epsilon_s \mathbb{1}\{a_s = 1\} \\
&= \lim_{t\to\infty} \left( \sum_{s=1}^{t} \mathbf{x}_s' \mathbf{x}_s'^\top \mathbb{1}\{a_s = 1\} \right)^{-1} \left( \sum_{s=1}^{t} \mathbf{x}_s' \mathbf{x}_s^\top \mathbb{1}\{a_s = 1\} \right) \boldsymbol{\theta}_*^{(1)}
\end{aligned}$$

$\qquad\square$

## C Proofs for Section 4

### C.1 Proof of Theorem 4.1

**Theorem C.1.** *Algorithm 4 with* $\eta = \sqrt{\frac{\log(|\mathcal{E}|)}{T\lambda^2|\mathcal{E}|}}$, $\gamma = 2\eta\lambda|\mathcal{E}|$, *and* $\varepsilon = \left( \frac{d\sigma \log T}{T} \right)^{1/(d+2)}$ *incurs expected strategic regret:*

$$\mathbb{E}[\text{Reg}(T)] \leq 6T^{(d+1)/(d+2)} (d\sigma \log T)^{1/(d+2)} = \widetilde{\mathcal{O}}\left( T^{(d+1)/(d+2)} \right).$$

*Proof.* Let $a_{t,e}$ correspond to the action chosen by a grid point $e \in \mathcal{E}$. We simplify notation to $a_t = a_{t,e_t}$ to be the action chosen by the sampled grid point $e_t$ at round $t$. For the purposes of the analysis, we also define $\ell_t(e) = 1 + \lambda - r_t(a_{t,e_t})$.

We first analyze the difference between the loss of the algorithm and the best-fixed point on the grid $e^*$, i.e.,

$$\mathbb{E}[\mathrm{Reg}^\varepsilon(T)] = \max_{e^* \in \mathcal{E}} \mathbb{E}\left[\sum_{t \in [T]} r_t(a_{t,e^*})\right] - \mathbb{E}\left[\sum_{t \in [T]} r_t(a_t)\right]$$

$$= \mathbb{E}\left[\sum_{t \in [T]} \ell_t(e_t)\right] - \min_{e^* \in \mathcal{E}} \mathbb{E}\left[\sum_{t \in [T]} \ell_t(e^*)\right]$$

where the equivalence between working with $\ell_t(\cdot)$ as opposed to $r_t(\cdot)$ holds because $\ell_t(\cdot)$ are just a shift from $r_t(\cdot)$ that is the same across all rounds and experts. For the regret of the algorithm, we show that:

$$\mathbb{E}[\mathrm{Reg}^\varepsilon(T)] = O\left(T \cdot d \cdot \left(\frac{1}{\varepsilon}\right)^{2d} \cdot \log\left(\frac{1}{\varepsilon}\right)\right) \tag{4}$$

We define the "good" event as the event that the reward is in $[0, 1]$ for every round $t$: $\mathcal{C} = \{r_t \in [0, 1], \forall t \in [T]\}$. Note that this depends on the noise of the round $\varepsilon_t$. We will call the complement of the "good" event, the "bad" event $\neg\mathcal{C}$. The regret of the algorithm depends on both $\mathcal{C}$ and $\neg\mathcal{C}$ as follows:

$$\mathbb{E}[\mathrm{Reg}^\varepsilon(T)] = \mathbb{E}[\mathrm{Reg}^\varepsilon(T)|\mathcal{C}] \cdot \mathrm{Pr}[\mathcal{C}] + \mathbb{E}[\mathrm{Reg}^\varepsilon(T)|\neg\mathcal{C}] \cdot \mathrm{Pr}[\neg\mathcal{C}] \le \mathbb{E}[\mathrm{Reg}^\varepsilon(T)|\mathcal{C}] + T \cdot \mathrm{Pr}[\neg\mathcal{C}] \tag{5}$$

where the inequality is due to the fact that $\mathrm{Pr}[\mathcal{C}] \le 1$ and that in the worst case, the algorithm must pick up a loss of 1 at each round.

We now upper bound the probability with which the bad event happens.

$$\mathrm{Pr}[\neg\mathcal{C}] = \mathrm{Pr}[\exists t : r_t \notin [0,1]] \le \sum_{t \in [T]} \mathrm{Pr}[r_t \notin [0,1]] \qquad \text{(union bound)}$$

$$\le \sum_{t \in [T]} \mathrm{Pr}[|\varepsilon_t| \ge \lambda] \le 2\exp(-\lambda^2/\sigma^2) \cdot T \le \frac{2}{T} \qquad \text{(substituting } \lambda\text{)}$$

Plugging $\mathrm{Pr}[\neg\mathcal{C}]$ to Equation (5) we get:

$$\mathbb{E}[\mathrm{Reg}^\varepsilon(T)] \le \mathbb{E}[\mathrm{Reg}^\varepsilon(T)|\mathcal{C}] + 2 \tag{6}$$

So for the remainder of the proof we will condition on the clean event $\mathcal{C}$ and compute $\mathbb{E}[\mathrm{Reg}^\varepsilon(T)|\mathcal{C}]$. Conditioning on $\mathcal{C}$ means that $1 + \lambda - r_t(a) \in [0, \lambda]$, where $\lambda = \sigma\sqrt{\log T}$.

We first compute the first and the second moments of estimator $\widehat{\ell}_t(\cdot)$. For the first moment:

$$\mathbb{E}\left[\widehat{\ell}_t(e)\right] = \sum_{e' \in \mathcal{E}} q_t(e') \cdot \frac{\ell_t(e) \cdot \mathbb{1}\{e = e'\}}{q_t(e)} = \ell_t(e) \tag{7}$$

For the second moment:

$$\mathbb{E}\left[\widehat{\ell}_t^2(e)\right] = \sum_{e' \in \mathcal{E}} q_t(e') \frac{\ell_t^2(e) \cdot \mathbb{1}\{e = e'\}}{q_t^2(e)} = \frac{\ell_t^2(e)}{q_t(e)} \le \frac{\lambda^2}{q_t(e)} \tag{8}$$

where for the first inequality, we have used the fact that $\ell_t(e) \le \lambda$ (since we conditioned on $\mathcal{C}$) and the last one is due to the fact that $q_t(e) \ge \gamma/|\mathcal{E}|$.

We define the weight assigned to grid point $e \in \mathcal{E}$ at round $t$ as: $w_t(e) = w_{t-1}(e) \cdot \exp(-\eta\widehat{\ell}_t(e))$ and $w_0(e) = 1, \forall e \in \mathcal{E}$. Let $W_t = \sum_{e \in \mathcal{E}} w_t(e)$ be the potential function. Then,

$$W_0 = \sum_{e \in \mathcal{E}} w_0(e) = |\mathcal{E}| \tag{9}$$

Using $e^*$ to denote the best-fixed policy in hindsight, we have:

$$W_T = \sum_{e \in \mathcal{E}} w_T(e) \ge w_T(e^*) = \exp\left(-\eta \sum_{t \in [T]} \widehat{\ell}_t(e^*)\right) \tag{10}$$

We next analyze how much the potential changes per-round:

$$\log\left(\frac{W_{t+1}}{W_t}\right) = \log\left(\frac{\sum_{e\in\mathcal{E}} w_t(e)\exp\left(-\eta\widehat{\ell}_t(e)\right)}{W_t}\right) = \log\left(\sum_{e\in\mathcal{E}} p_t(e)\exp\left(-\eta\widehat{\ell}_t(e)\right)\right)$$

$$\leq \log\left(\sum_{e\in\mathcal{E}} p_t(e)\cdot\left(1 - \eta\widehat{\ell}_t(e) + \eta^2\widehat{\ell}_t^2(e)\right)\right) \qquad (e^{-x} \leq 1 - x + x^2, x > 0)$$

$$= \log\left(1 - \eta\sum_{e\in\mathcal{E}} p_t(e)\widehat{\ell}_t(e) + \eta^2\sum_{e\in\mathcal{E}} p_t(e)\widehat{\ell}_t^2(e)\right) \qquad (\textstyle\sum_{e\in\mathcal{E}} p_t(e) = 1)$$

$$\leq -\eta\sum_{e\in\mathcal{E}} p_t(e)\widehat{\ell}_t(e) + \eta^2\sum_{e\in\mathcal{E}} p_t(e)\widehat{\ell}_t^2(e)$$

$$= -\eta\sum_{e\in\mathcal{E}}\frac{q_t(e) - \gamma/|\mathcal{E}|}{(1-\gamma)}\widehat{\ell}_t(e) + \eta^2\sum_{e\in\mathcal{E}}\frac{q_t(e) - \gamma/|\mathcal{E}|}{(1-\gamma)}\widehat{\ell}_t^2(e)$$

$$\leq -\eta\sum_{e\in\mathcal{E}}\frac{q_t(e) - \gamma/|\mathcal{E}|}{(1-\gamma)}\widehat{\ell}_t(e) + \eta^2\sum_{e\in\mathcal{E}}\frac{q_t(e)}{(1-\gamma)}\widehat{\ell}_t^2(e) \tag{11}$$

where the second inequality is due to the fact that $\log x \leq x - 1$ for $x \geq 0$. In order for this inequality to hold we need to verify that:

$$1 - \eta\sum_{e\in\mathcal{E}} p_t(e)\widehat{\ell}_t(e) + \eta^2\sum_{e\in\mathcal{E}} p_t(e)\widehat{\ell}_t^2(e) \geq 0,$$

or equivalently, that:

$$1 - \eta\sum_{e\in\mathcal{E}} p_t(e)\widehat{\ell}_t(e) \geq 0 \tag{12}$$

We do so after we explain how to tune $\eta$ and $\gamma$.

We return to Equation (11); summing up for all rounds $t \in [T]$ in Equation (11) we get:

$$\log\left(\frac{W_T}{W_0}\right) \leq -\eta\sum_{t\in[T]}\sum_{e\in\mathcal{E}}\frac{q_t(e) - \gamma/|\mathcal{E}|}{(1-\gamma)}\widehat{\ell}_t(e) + \eta^2\sum_{t\in[T]}\sum_{e\in\mathcal{E}}\frac{q_t(e)}{(1-\gamma)}\widehat{\ell}_t^2(e) \tag{13}$$

Using Equation (9) and Equation (10) we have that: $\log(W_T/W_0) \geq -\eta\sum_{t\in[T]}\widehat{\ell}_t(e^*) - \log|\mathcal{E}|$. Combining this with the upper bound on $\log(W_T/W_0)$ from Equation (13) and multiplying both sides by $(1-\gamma)/\eta$ we get:

$$\sum_{t\in[T]}\sum_{e\in\mathcal{E}}\left(q_t(e) - \frac{\gamma}{|\mathcal{E}|}\right)\widehat{\ell}_t(e) - (1-\gamma)\sum_{t\in[T]}\widehat{\ell}_t(e^*) \leq \eta\sum_{t\in[T]}\sum_{e\in\mathcal{E}} q_t(e)\widehat{\ell}_t^2(e) + (1-\gamma)\frac{\log(|\mathcal{E}|)}{\eta}$$

We can slightly relax the right hand side using the fact that $\gamma < 1$ and get:

$$\sum_{t\in[T]}\sum_{e\in\mathcal{E}}\left(q_t(e) - \frac{\gamma}{|\mathcal{E}|}\right)\widehat{\ell}_t(e) - (1-\gamma)\sum_{t\in[T]}\widehat{\ell}_t(e^*) \leq \eta\sum_{t\in[T]}\sum_{e\in\mathcal{E}} q_t(e)\widehat{\ell}_t^2(e) + \frac{\log(|\mathcal{E}|)}{\eta}$$

Taking expectations (wrt the draw of the algorithm) on both sides of the above expression and using our derivations for the first and second moment (Equation (7) and Equation (8) respectively) we get:

$$\sum_{t\in[T]}\sum_{e\in\mathcal{E}}\left(q_t(e) - \frac{\gamma}{|\mathcal{E}|}\right)\ell_t(e) - (1-\gamma)\sum_{t\in[T]}\ell_t(e^*) \leq \eta\sum_{t\in[T]}\sum_{e\in\mathcal{E}} q_t(e)\frac{\lambda^2}{q_t(e)} + \frac{\log(|\mathcal{E}|)}{\eta}$$

Using the fact that $\ell_t(\cdot) \in [0,\lambda]$ the above becomes:

$$\mathbb{E}\left[\mathrm{Reg}^\varepsilon(T)|\mathcal{C}\right] = \sum_{t\in[T]}\sum_{e\in\mathcal{E}} q_t(e)\ell_t(e) - \sum_{t\in[T]}\ell_t(e^*) \leq \eta T\lambda^2|\mathcal{E}| + \frac{\log(|\mathcal{E}|)}{\eta} + \gamma T$$

Tuning $\eta = \sqrt{\frac{\log(|\mathcal{E}|)}{T\lambda^2|\mathcal{E}|}}$ and $\gamma = 2\eta\lambda|\mathcal{E}|$, we get that:

$$\mathbb{E}\left[\text{Reg}^\varepsilon(T)|\mathcal{C}\right] \leq 3\sqrt{T|\mathcal{E}|\lambda^2\log(|\mathcal{E}|)} = 3\sqrt{T|\mathcal{E}|\sigma\log(T)\log(|\mathcal{E}|)} \tag{14}$$

Before we proceed to bounding the discretization error that we incur by playing policies only on the grid, we verify that Equation (12) holds for the chosen $\eta$ and $\gamma$ parameters. Note that when $\widehat{\ell}_t(e) = 0$, then Equation (12) holds. So we focus on the case where $\widehat{\ell}_t(e) = \ell_t(e)/q_t(e)$.

$$\eta\sum_{e\in\mathcal{E}}p_t(e)\frac{\ell_t(e)}{q_t(e)} \leq \eta\sum_{e\in\mathcal{E}}p_t(e)\frac{\ell_t(e)\cdot|\mathcal{E}|}{\gamma} \leq \eta\sum_{e\in\mathcal{E}}p_t(e)\frac{\lambda\cdot|\mathcal{E}|}{\gamma} = \eta\frac{\lambda|\mathcal{E}|}{\gamma} = \frac{1}{2}$$

where the first inequality is due to the fact that $q_t(e) \geq \gamma/|\mathcal{E}|, \forall e \in \mathcal{E}$, the second is because $\ell_t(e) \leq \lambda$, the first equality is because $\sum_{e\in\mathcal{E}}p_t(e) = 1$, and the last equality is because of the values that we chose for parameters $\eta$ and $\gamma$.

The final step in proving the theorem is to bound the strategic discretization error that we incur because our algorithm only chooses policies on the grid, while $\boldsymbol{\theta}^{(1)}, \boldsymbol{\theta}^{(0)}$ (and hence, the actual optimal policy) may not correspond to any grid point. Let $a_t^*$ correspond to the action chosen by the optimal policy.

$$SDE(T) = \sum_{t\in[T]}\mathbb{E}\left[r_t(a_t^*)\right] - \sum_{t\in[T]}\mathbb{E}\left[r_t(a_{t,e^*})\right] = \sum_{t\in[T]}\left\langle\boldsymbol{\theta}^{(a_t^*)} - \boldsymbol{\theta}^{(a_{t,e^*})}, \mathbf{x}_t\right\rangle$$

We separate the $T$ rounds into 3 groups: in group $G_1$, we have rounds $t \in [T]$ such that $a_t^* = a_{t,e^*}$. In group $G_2$, we have rounds $t \in [T]$ such that $a_t^* = 0$ but $a_{t,e^*} = 1$. In group $G_3$, we have rounds $t \in [T]$, such that $a_t^* = 1$ but $a_{t,e^*} = 0$. With these groups in mind, one can rewrite the above equation as:

$$SDE(T) = \sum_{t\in G_1}\left\langle\boldsymbol{\theta}^{(a_t^*)} - \boldsymbol{\theta}^{(a_{t,e^*})}, \mathbf{x}_t\right\rangle + \sum_{t\in G_2}\left\langle\boldsymbol{\theta}^{(a_t^*)} - \boldsymbol{\theta}^{(a_{t,e^*})}, \mathbf{x}_t\right\rangle + \sum_{t\in G_3}\left\langle\boldsymbol{\theta}^{(a_t^*)} - \boldsymbol{\theta}^{(a_{t,e^*})}, \mathbf{x}_t\right\rangle$$

For all the rounds in $G_1$, the strategic discretization error is equal to 0. Hence the strategic discretization error becomes:

$$SDE(T) = \underbrace{\sum_{t\in G_2}\left\langle\boldsymbol{\theta}^{(a_t^*)} - \boldsymbol{\theta}^{(a_{t,e^*})}, \mathbf{x}_t\right\rangle}_{SDE(G_2)} + \underbrace{\sum_{t\in G_3}\left\langle\boldsymbol{\theta}^{(a_t^*)} - \boldsymbol{\theta}^{(a_{t,e^*})}, \mathbf{x}_t\right\rangle}_{SDE(G_3)} \tag{15}$$

We first analyze $SDE(G_2)$:

$$SDE(G_2) = \sum_{t\in G_2}\left\langle\boldsymbol{\theta}^{(0)} - \boldsymbol{\theta}^{(1)}, \mathbf{x}_t\right\rangle$$

Let us denote by $\widehat{\boldsymbol{\theta}}^{(1)}$ and $\widehat{\boldsymbol{\theta}}^{(0)}$ the points such that $e^* = \widehat{\boldsymbol{\theta}}^{(1)} - \widehat{\boldsymbol{\theta}}^{(0)}$. Adding and subtracting $\langle e^*, \mathbf{x}_t\rangle$ in the above, we get:

$$SDE(G_2) = \sum_{t\in G_2}\left(\left\langle\boldsymbol{\theta}^{(0)} - \widehat{\boldsymbol{\theta}}^{(0)}, \mathbf{x}_t\right\rangle + \left\langle\widehat{\boldsymbol{\theta}}^{(1)} - \boldsymbol{\theta}^{(1)}, \mathbf{x}_t\right\rangle + \left\langle\widehat{\boldsymbol{\theta}}^{(0)} - \widehat{\boldsymbol{\theta}}^{(1)}, \mathbf{x}_t\right\rangle\right)$$

$$\leq \sum_{t\in G_2}\left(\left|\left\langle\boldsymbol{\theta}^{(0)} - \widehat{\boldsymbol{\theta}}^{(0)}, \mathbf{x}_t\right\rangle\right| + \left|\left\langle\widehat{\boldsymbol{\theta}}^{(1)} - \boldsymbol{\theta}^{(1)}, \mathbf{x}_t\right\rangle\right| + \left\langle\widehat{\boldsymbol{\theta}}^{(0)} - \widehat{\boldsymbol{\theta}}^{(1)}, \mathbf{x}_t\right\rangle\right) \qquad (x \leq |x|)$$

$$\leq 2\varepsilon T + \sum_{t\in G_2}\underbrace{\left\langle\widehat{\boldsymbol{\theta}}^{(0)} - \widehat{\boldsymbol{\theta}}^{(1)}, \mathbf{x}_t\right\rangle}_{Q_t} \qquad \text{(Cauchy-Schwarz)}$$

Finally, we show that $Q_t \leq 0$. For the rounds where $a_{t,e^*} = 1$ but $a_t^* = 0$, it can be the case that $\mathbf{x}_t \neq \mathbf{x}_t'$ (as the agents only strategize in order to get assigned action 1. But since $a_{t,e^*} = 1$, then from the algorithm:

$$\left\langle\widehat{\boldsymbol{\theta}}^{(1)} - \widehat{\boldsymbol{\theta}}^{(0)}, \mathbf{x}_t'\right\rangle \geq \delta\|e^*\| \Leftrightarrow \left\langle\widehat{\boldsymbol{\theta}}^{(0)} - \widehat{\boldsymbol{\theta}}^{(1)}, \mathbf{x}_t'\right\rangle \leq -\delta\|e^*\| \tag{16}$$

Adding and subtracting $\mathbf{x}'_t$ from quantity $Q_t$, we have:

$$Q_t = \left\langle \widehat{\boldsymbol{\theta}}^{(0)} - \widehat{\boldsymbol{\theta}}^{(1)}, \mathbf{x}_t - \mathbf{x}'_t \right\rangle + \left\langle \widehat{\boldsymbol{\theta}}^{(0)} - \widehat{\boldsymbol{\theta}}^{(1)}, \mathbf{x}'_t \right\rangle$$

$$\leq \left\langle \widehat{\boldsymbol{\theta}}^{(0)} - \widehat{\boldsymbol{\theta}}^{(1)}, \mathbf{x}_t - \mathbf{x}'_t \right\rangle - \delta \|e^*\| \qquad \text{(Equation (16))}$$

$$\leq \left\| \widehat{\boldsymbol{\theta}}^{(0)} - \widehat{\boldsymbol{\theta}}^{(1)} \right\| \cdot \|\mathbf{x}_t - \mathbf{x}'_t\| - \delta \|e^*\| \qquad \text{(Cauchy-Schwarz)}$$

$$\leq \|e^*\| \cdot \delta - \delta \|e^*\|.$$

As a result:

$$SDE(G_2) \leq 2\varepsilon T \qquad (17)$$

Moving on to the analysis of $SDE(G_3)$:

$$SDE(G_3) = \sum_{t \in G_3} \left\langle \boldsymbol{\theta}^{(0)} - \boldsymbol{\theta}^{(1)}, \mathbf{x}_t \right\rangle$$

Again, we use $\widehat{\boldsymbol{\theta}}^{(1)}$ and $\widehat{\boldsymbol{\theta}}^{(0)}$ the points that $e^* = \widehat{\boldsymbol{\theta}}^{(1)} - \widehat{\boldsymbol{\theta}}^{(0)}$. Adding and subtracting $\langle e^*, \mathbf{x}_t \rangle$ and following the same derivations as in $SDE(G_3)$, we have that:

$$SDE(G_3) \leq 2\varepsilon T + \sum_{t \in G_3} \underbrace{\left\langle \widehat{\boldsymbol{\theta}}^{(1)} - \widehat{\boldsymbol{\theta}}^{(0)}, \mathbf{x}_t \right\rangle}_{Q_t} \qquad (18)$$

Since $a_{t,e^*} = 0$, then it must have been the case that $\mathbf{x}'_t = \mathbf{x}_t$; this is because the agent would not spend effort to strategize if they would still be assigned action 0. For this reason, it must be that $Q_t \leq 0$.

Combining the upper bounds for $SDE(G_2)$ and $SDE(G_3)$ in Equation (15), we have that $SDE(T) \leq 4\varepsilon T$.

Putting everything together, we have that the regret is comprised by the regret incurred on the discretized grid and the strategic discretization error, i.e.,

$$\mathbb{E}[\text{Reg}(T)] \leq 3\sqrt{T|\mathcal{E}|\sigma \log(T) \log(|\mathcal{E}|)} + 4\varepsilon T = 3\sqrt{Td\left(\frac{1}{\varepsilon}\right)^d \sigma \log(T) \log(1/\varepsilon)} + 4\varepsilon T$$

Tuning $\varepsilon = \left(\frac{d\sigma \log T}{T}\right)^{1/(d+2)}$ we get that the regret is:

$$\mathbb{E}[\text{Reg}(T)] \leq 6T^{(d+1)/(d+2)}(d\sigma \log T)^{1/(d+2)} = \widetilde{\mathcal{O}}\left(T^{(d+1)/(d+2)}\right).$$

$\qquad\qquad\qquad\qquad\qquad\qquad\qquad\qquad\qquad\qquad\qquad\qquad\qquad\qquad\qquad\qquad\qquad\square$

## D  Extension to trembling hand best-response

Observe that when lazy tiebreaking (Definition 2.1), if agent $t$ modifies their context they modify it by an amount $\delta_L$ such that

$$\delta_{L,t} := \min_{0 \leq \eta \leq \delta} \eta$$
$$\text{s.t. } \pi_t(\mathbf{x}'_t) = 1$$
$$\|\mathbf{x}'_t - \mathbf{x}_t\|_2 = \eta.$$

We define $\gamma$-*trembling hand* tiebreaking as $\delta_{TH,t} = \delta_{L,t} + \alpha_t$, where $\alpha_t \in [0, \min\{\delta - \delta_{L,t}, \gamma\}]$ may be chosen arbitrarily. Our results in Section 3 may be extended to trembling hand tiebreaking by considering the following redefinition of a clean point:

**Condition D.1** (Sufficient condition for $\mathbf{x}' = \mathbf{x}$). *Given a shifted linear policy parameterized by* $\boldsymbol{\beta}^{(1)} \in \mathbb{R}^d$, *we say that a context* $\mathbf{x}'$ *is* clean *if* $\langle \boldsymbol{\beta}^{(1)}, \mathbf{x}' \rangle > (\delta + \gamma)\|\boldsymbol{\beta}^{(1)}\|_2 + r_0$.

No further changes are required. This will result in a slightly worse constant in Theorem 3.3 (i.e. all instances of $\delta$ will be replaced by $\delta + \gamma$). Our algorithms and results in Section 4 do not change. Our definition of trembling hand best-response is similar in spirit to the $\epsilon$-best-response in Haghtalab et al. [26]. Specifically, Haghtalab et al. [26] study a Stackelberg game setting in which the follower best-responds $\epsilon$-optimally. In our trembling hand setting, the strategic agent can also be thought of as $\epsilon$-best responding (using the language of [26]), although it is important to note that an $\epsilon$-best response for the agent in our setting will cause the agent to only strategize *more* than necessary.

