# OpenReview forum: "Strategic Apple Tasting"
_NeurIPS.cc/2023/Conference — NeurIPS 2023 poster_

### Official Review · Reviewer_qFGe · 2023-06-21

**Soundness:** 3 good
**Presentation:** 3 good
**Contribution:** 3 good
**Rating:** 6
**Confidence:** 4

**Summary:**

The paper studies an online learning problem with incentives. In particular, in the model studied in the paper there is a principal who has to take decisions on a sequence of $T$ (different) agents. Each agent has a context and they can strategically disclose it (truthfully or untruthfully) to the principal in order to induce them to take favorable decisions. The key feature of the model studied in the paper is that the feedback is one-sided, meaning that the principal only observes a feedback when they take a positive decision on the agent. The paper first proposes an algorithm that achieves $\tilde O(\sqrt{T})$ strategic regret (a stronger notion than Stackelberg regret), when the the contexts are selected stochastically. Then, the paper shows how to deal with adversarially-chosen contexts, by providing an algorithm with $\tilde O(T^{(d+1)/(d+2)})$ strategic regret.

**Strengths:**

ORIGINALITY

The paper studies, for the first time to the best of my knowledge, an online learning problem in which both incentives and one-sided feedback are involved. This is a relevant combination present in may application domains.

QUALITY

The paper does a good job in addressing several aspects of the online learning problem under study. In particular, the paper studies both the case in which contexts are selected stochastically and the one in which they are chosen adversarially.

CLARITY

The paper is well written and easy to follow.

SIGNIFICANCE

I believe that the problem studied in the paper is of interest several people spanning different communities, from online learning to algorithmic game theory.

**Weaknesses:**

ORIGINALITY

The techniques used in the paper are not groundbreaking, but they are rather adaptations of well-known and widely-used techniques in online learning.

QUALITY

While the literature revised in the paper is quite extensive as far as work on online learning with incentives are concerned, I think the paper is missing some very related works in the literature at the interface between algorithmic game theory and online learning. In particular, the following lines of research seem strongly related to the paper:
- Repeated Stackelberg games: Balcan, Maria-Florina, et al. "Commitment without regrets: Online learning in stackelberg security games." Proceedings of the sixteenth ACM conference on economics and computation. 2015. (see also some subsequent works extending the paper)
- Online Bayesian persuasion: Castiglioni, Matteo, et al. "Online bayesian persuasion." Advances in Neural Information Processing Systems 33 (2020): 16188-16198. + Castiglioni, Matteo, et al. "Multi-receiver online bayesian persuasion." International Conference on Machine Learning. PMLR, 2021. (these works extend those on repeated Stackelberg games to the Bayesian persuasion framework)
- Online learning in principal-agent problems: Zhu, Banghua, et al. "The Sample Complexity of Online Contract Design." arXiv preprint arXiv:2211.05732 (2022). (a recent preprint that indeed seems very related to the present paper)
I would have expected a discussion on these works in the paper. Of course, the papers that I cited above are only some representative of such a line of research, and I am happy to provide more references if the authors need them.

CLARITY

No weaknesses to report.

SIGNIFICANCE

No weaknesses to report.

**Questions:**

1) I believe that the model studied in the paper is closely connected with one introduced in a very recent preprint (Bernasconi, Martino, et al. "Optimal Rates and Efficient Algorithms for Online Bayesian Persuasion." arXiv preprint arXiv:2303.01296 (2023).). In that paper, the authors study a model with type reporting that looks very similar to the one in the present paper. Am I right?
2) The regret bound in Theorem 4.1 is very similar to a regret bound in (Zhu, Banghua, et al. "The Sample Complexity of Online Contract Design." arXiv preprint arXiv:2211.05732 (2022).). Can you discuss the differences between the approach in that paper and your approach?
3) Algorithm 3 requires exponential running time in the worst case. Are there any computational hardness results that can be applied to your problem?

**Limitations:**

No limitations.

---

> ### Author Rebuttal · Authors · 2023-08-09
>
> Thank you for taking the time to review our work. Please find our answers to your comments/questions below.
>
> [*While the literature revised in the paper is quite extensive as far as work on online learning with incentives are concerned, I think the paper is missing some very related works in the literature at the interface between algorithmic game theory and online learning. In particular, the following lines of research seem strongly related to the paper…*]
>
> While we agree that the settings you mention are related in the sense that all of these settings (including ours) are various instantiations of Stackelberg games, we did not originally include them as we believe that they are not as strongly related to our work as the literature we already cited. However, we are happy to include them in an expanded related work section and we highlight below the major differences and similarities with our paper.
>
> Repeated Stackelberg games: In this literature, the principal (leader) commits to a mixed strategy over a finite set of actions, and the agent (follower) best-responds by playing an action from a finite set of best-responses. Note that unlike in our setting, both the principal’s and agent’s payoffs can be represented by matrices. In contrast, in our setting the principal commits to a pure strategy from a continuous set of actions, and the agent best-responds by playing an action from a continuous set.
>
> Online Bayesian persuasion: In this literature, the principal (sender) commits to a “signaling policy” (a random mapping from “states of the world” to receiver actions) and the agent (receiver) performs a posterior update on the state based on the principal’s signal, then takes an action from a (usually finite) set. In both this setting and ours, the principal’s action is a policy. However in our setting the policy is a linear decision rule, whereas in the Bayesian persuasion setting, the policy is a set of conditional probabilities which form an “incentive compatible” signaling policy. This difference in the policy space for the principal typically leads to different algorithmic ideas being used in the two settings.
>
> Online learning in principal-agent problems: Strategic learning problems are indeed an instance of online learning in principal-agent problems. Since you mention online contract design as a particular example, we will highlight how our work differs from this area. In contract design, the principal commits to a contract (a mapping from “outcomes” to agent payoffs). The agent then takes an action, which affects the outcome. In particular, they take the action which maximizes their expected payoff, subject to some cost of taking the action. The goal of the principal is to design a contract such that their own expected payoff is maximized. While the settings are indeed similar, there are several key differences. First, in online contract design the principal always observes the outcome, whereas in our setting the principal only observes the reward if a positive decision is made. Second, the form of the agent’s best response is different, which leads to different agent behavior and, as a result, different online algorithms for the principal (see below for more details).
>
>
> [*I believe that the model studied in the paper is closely connected with one introduced in a very recent preprint (Bernasconi, Martino, et al. "Optimal Rates and Efficient Algorithms for Online Bayesian Persuasion." arXiv preprint arXiv:2303.01296 (2023).). In that paper, the authors study a model with type reporting that looks very similar to the one in the present paper. Am I right?*]
>
> In type reporting, the sender (principal) asks each receiver (agent) to select a signaling scheme, and the signaling schemes are designed in a way such that each receiver is incentivized to select the signaling scheme corresponding to their (true) type. In our setting, the agent’s “type” may be thought of as their context. However unlike the online Bayesian persuasion setting, in our setting the agent has an incentive to strategically misreport their type. Another (less important) difference between our setting and this one is that in our setting the agent’s type is from a continuous space, but the type space is discrete in the setting of Bernasconi et al. We will add these connections in a revision.
>
> [*The regret bound in Theorem 4.1 is very similar to a regret bound in (Zhu, Banghua, et al. "The Sample Complexity of Online Contract Design." arXiv preprint arXiv:2211.05732 (2022).). Can you discuss the differences between the approach in that paper and your approach?*]
>
> The two algorithms are similar, as both are running multi-armed bandit algorithms (in our case EXP3, in their case UCB) over a discretization of a continuous action space. As a result, our novelty in this section is the analysis of the algorithm in the strategic setting we consider. (Note that the analysis of our algorithm differs significantly from theirs; ours relies on bounding the “strategic discretization error”, whereas theirs uses a covering argument to bound discretization error.)
>
> [*Algorithm 3 requires exponential running time in the worst case. Are there any computational hardness results that can be applied to your problem?*]
>
> We are not aware of any such hardness results. As we mention in the conclusion, an exciting direction for future research is to design polynomial-time algorithms (or prove hardness results) for the adversarially-chosen context setting.

---

> > ### Comment · Reviewer_qFGe · 2023-08-20
> >
> > I would like to thank the Authors for their response, they satisfactorily answered all of my questions. As a result, I am sticking to my (positive) score.

---

### Official Review · Reviewer_Mu3N · 2023-06-25

**Soundness:** 3 good
**Presentation:** 3 good
**Contribution:** 3 good
**Rating:** 6
**Confidence:** 3

**Summary:**

The paper studies strategic apple tasting settings. This setting involves decision making that assigns decisions to agents who have incentives to strategically modify their input (context), and the decision maker only receives apple tasting (one-sided) feedback, where it only receives feedback for positively assigned decisions. The authors formulate this setting as an online learning problem with apple-tasting feedback, where a principal makes decisions about a sequence of T agents, with the goal of achieving sublinear strategic regret. Under this problem formulation, the authors proposed a learning algorithm (Algorithm 1) based on greedy, shifted linear policy, and parameter updates based on clean contexts. The author shows that the algorithm achieves O(\sqrt{T}) strategic regret when the sequence when the agent contexts are generated stochastically. Then, the authors extendes to the situation where T is small or unknown and proposed another algorithm (Algorithm 2) under this scenario. Finally, the authors relax the assumption and provide an algorithm (Algorithm 3) under adversarially generated agents. These results apply to the more general setting of bandit feedback, under slight modifications to the algorithms.

**Strengths:**

- I like that the paper is very well written and easy to follow
- The studied one-sided feedback settings with strategic agents are interesting and often occurs in real-world situations.
- The problem is clearly formulated. For all proposed algorithms, the authors provided performance guarantees which are clearly explained and supported by proofs.


**Weaknesses:**

It is slightly hard for me to make a connection between assumption of the agent's strategic modification to their context (with effort budget \delta mentioned in definition 2.1), and the motivated examples such as hiring and lending.

**Questions:**

- As in the above section, can the authors shed some light on the strategic modification effort budget?
- Line 212: The authors mentioned "This is the type of strategizing we want to incentivize." I didn't understand why the decision maker would like to incentivize this behavior?
- Algorithm 1 line 4, "... and data S_t", do you mean "data D_t"?
- Can the authors move algorithm 4 (which is part of algorithm 2) into the main text?

**Limitations:**

I see that the authors listed a few related future directions in the conclusion. No negative societal impact observed.

---

> ### Author Rebuttal · Authors · 2023-08-09
>
> Thank you for taking the time to review our work. Please find our responses to your questions and comments below.
>
> [*As in the above section, can the authors shed some light on the strategic modification effort budget?*]
>
> Consider a lending setting in which an applicant (strategic agent) wishes to obtain a loan from a bank (principal). Each agent is described by some context (e.g., number of lines of existing credit, credit history, current income, etc.), which may be strategically modified by the agent. Using the (possibly modified) context, the goal of the bank is to decide whether or not to give the applicant a loan. If the loan is given to the applicant, the bank observes some reward (e.g. whether or not the loan was paid back on time, amount of interest accrued, etc.). Otherwise if they reject the applicant, they receive no signal about their decision.
>
> Specifically regarding the effort budget, it may be viewed as a hard constraint on the amount an individual is able to strategically modify their original context. In settings such as lending, this budget is analogous to “hard”/”strict” time or monetary constraints that agents may have when modifying their context. For a more concrete example, a loan applicant may only have several hours every day during which he/she may prepare for their loan application (due to other responsibilities). In this case, the amount they can strategize is limited by this time constraint.
>
> [*Line 212: The authors mentioned "This is the type of strategizing we want to incentivize." I didn't understand why the decision maker would like to incentivize this behavior?*]
>
> By “this is the type of strategizing we want to incentivize”, we just meant that we want these particular agents to strategize so that we can assign the positive action to them (since this is what maximizes the principal’s utility), as we shifted the decision boundary to account for the strategic behavior of the agents who should not receive the positive action. We will add a clarifying sentence in our revision.
>
> [*Algorithm 1 line 4, "... and data S_t", do you mean "data D_t"?*]
>
> Yes, thank you for pointing this out.
>
> [*Can the authors move algorithm 4 (which is part of algorithm 2) into the main text?*]
>
> Yes – we will do so in the revision.

---

### Official Review · Reviewer_6Gq7 · 2023-06-30

**Soundness:** 3 good
**Presentation:** 3 good
**Contribution:** 2 fair
**Rating:** 5
**Confidence:** 2

**Summary:**

The paper considers contextual bandit problem in which the users can strategically modify their context for its own sake.
It provides sublinear regret algorithms by exploiting the budgeted strategic structure of the agents, against stochastically chosen agents.
It further obtains sublinear regret algorithms against adversarially chosen agents by considering a variant of EXP3.

**Strengths:**

The paper studies interesting problem setting and makes reasonable contribution.

**Weaknesses:**

Although the analysis seems technically thorough, intuition behind the solution is pretty straightforward: incentivize truthful action for certain amount of times, estimate $\theta$, strictly separate strategically modified context by conservatively setting decision boundary based on the effort budget.
Also, the algorithm seems to heavily depend on the information on the agents' budget.
Detailed comments are in Questions.

**Questions:**

- What happens if the principal can also optimize over the choice of decision policy, i.e., not just using thresholded rule?
- In L254, even though a small fraction of agents may strategically modify their contexts, it seems fairly intuitive that the estimates would be inconsistent, so wonder why the authors think this to be a surprising fact - is it meant to be "far" from being consistent, not just inconsistent?
- L82 consists a lot of citations. It looks bad to me, at least parsing them/giving more contexts to each reference/removing part of them would be helpful. It's a bit distracting in its current form and gave me no information.
- It seems that the algorithm essentially needs to know the budget $\delta$ in advance. What happens if the algorithm has misspecified budget? Is the algorithm fairly robust?
- In L170, bandit feedback setting is said to be more challenging - why is it more challenging than the one-sided feedback? Also, the authors argue that all their algorithms can be made to be applicable to the bandit setting - yes it's good to know that, but I wonder if there's more general results, e.g., any reduction from one setting to the other, or if the authors have thought about any black-box reduction from one algorithm to another.
- I also wonder why the standard adversarial contextual bandit algorithm does not work, instead of Algorithm 1, possibly with some slight modifications.
- I'd like to ask on how the authors think about the case if the agents observe some noisy context about themselves.

Minor comments
- I found the presentation of Stackelberg regret and Proposition 2.4 a bit redundant, as it does not appear in the main paper but there.
- In Algorithm 1, what is data $S_t$ - it seems it's not updated at all?
- L290, typo: withing
- L298, duplicated threshold


**Limitations:**

Detailed comments are in Questions.

---

> ### Author Rebuttal · Authors · 2023-08-09
>
> Thank you for taking the time to review our work. Please find out responses to your comments and questions below.
>
> [*What happens if the principal can also optimize over the choice of decision policy, i.e., not just using thresholded rule?*]
>
> This is an interesting question. While we are able to obtain no-strategic-regret by only considering linear decision policies, it may be possible in theory to obtain better rates if non-linear decision policies are deployed. However this appears challenging, as we point out in Section 3.1 “While our results do not rule out better strategic regret rates in d for more complicated algorithms (e.g., those which deploy non-linear policies), it is often unclear how strategic agents would behave in such settings, both in theory (Definition 2.1 would require agents to solve a non-convex optimization with potentially no closed-form solution) and in practice, making the analysis of such nonlinear policies difficult in strategic settings.”
>
> [*In L254, even though a small fraction of agents may strategically modify their contexts, it seems fairly intuitive that the estimates would be inconsistent, so wonder why the authors think this to be a surprising fact - is it meant to be "far" from being consistent, not just inconsistent?*]
>
> On line 253 we say that this phenomenon is unsurprising (not “surprising” as the reviewer mentions) - we include it because it motivates the study of “strategy-aware” algorithms.
>
> [*L82 consists a lot of citations. It looks bad to me, at least parsing them/giving more contexts to each reference/removing part of them would be helpful. It's a bit distracting in its current form and gave me no information.*]
>
> We chose to include the citations to show that there is a lot of work in this area which does not consider one-sided feedback. However we see your point, and will give more context to these references in an appendix.
>
> [*It seems that the algorithm essentially needs to know the budget $\delta$ in advance. What happens if the algorithm has misspecified budget? Is the algorithm fairly robust?*]
>
> Algorithm 1 is fairly robust to overestimates of the budget $\delta$, in the sense that (1) it will still produce a consistent estimate for $\theta^{(1)}$ (albeit at a rate which depends on the over-estimate instead of the actual value of $\delta$) and (2) it will incur a constant penalty in regret which is proportional to the amount of over-estimation. Algorithm 4 will also incur a constant penalty in regret which is proportional to the amount of misspecification (either over- or under-estimation). We will add a short discussion on this in the revision after the presentation of our bounds.
>
> [*In L170, bandit feedback setting is said to be more challenging - why is it more challenging than the one-sided feedback? Also, the authors argue that all their algorithms can be made to be applicable to the bandit setting - yes it's good to know that, but I wonder if there's more general results, e.g., any reduction from one setting to the other, or if the authors have thought about any black-box reduction from one algorithm to another.*]
>
> Bandit feedback is (slightly) more challenging in our setting, since an additional parameter ($\theta^{(0)}$) needs to be estimated. The algorithms for both feedback settings are more-or-less the same: the only difference is to estimate and use this additional parameter in the bandit feedback setting. We chose to present our results in terms of apple tasting feedback since that is the type of feedback present in our motivating examples.
>
> [*I also wonder why the standard adversarial contextual bandit algorithm does not work, instead of Algorithm 1, possibly with some slight modifications.*]
>
> We are not sure which algorithm you are referring to when you say “standard adversarial contextual bandit algorithm”. If you mean EXP4, this requires a set of (strategy-aware) experts for which we know the action they would recommend at each round. But note that the action per expert depends on the agent’s context, which is strategically changed as a response to the chosen expert. In other words, unless all agents are truthful with their contexts, we cannot infer the actions that all the experts would recommend per round (which is required for the EXP4 update).
>
> If you are referring to the more recent line of work on “adversarial contextual learning” (e.g., “Efficient Algorithms for Adversarial Contextual Learning” by Syrgkanis et al., “Improved Regret Bounds for Oracle-Based Adversarial Contextual Bandits” by Syrgkanis et al., “BISTRO” by Rakhlin & Sidharan), they require advanced knowledge of the distribution over contexts to give as input to their algorithms (an assumption we do not require). Moreover in the strategic setting, the distribution over contexts changes as a function of the algorithm deployed by a learner, since different algorithms will cause agents to strategize in different ways. Finally note that Algorithm 1 enjoys a better dependence on $T$ in the regret bound when compared to algorithms in this line of work ($T^{1/2}$ versus $T^{2/3}$ and $T^{3/4}$).
>
> [*I'd like to ask on how the authors think about the case if the agents observe some noisy context about themselves.*]
>
> If agents observe a noisy version of their context (instead of their true context), then a best-response similar to our “trembling hand tiebreaking” assumption (see Line 133 in the main body and Appendix D for more details) may be reasonable since in this case, an individual who strategizes may want to “play it safe” and “overshoot” the decision boundary by a bit to account for the fact that they do not have perfect knowledge of their context.
>
> Finally, thank you for pointing out the typos and suggestions. In particular, data $S_t$ is a typo - this should be data $D_t$. We will correct this and all others mentioned.

---

> > ### Comment · Reviewer_6Gq7 · 2023-08-15
> >
> > Thanks for your response. I have one more quick question that just comes to my mind.
> > Can you also discuss some connections to perturbation technique used in Learning in Stackelberg Games with Non-myopic Agents, EC'22 (Section 4.3), which refers back to Stochastic multiarmed bandits with unrestricted delay distributions, ICML'21?
> > I know that the setup is a bit different (contextual, one-sided, etc) but the objective of techniques seems related (to view the strategically modified feedback as a perturbed output of bandit problem).

---

> > > ### Author Response · Authors · 2023-08-15
> > >
> > > *Can you also discuss some connections to perturbation technique used in Learning in Stackelberg Games with Non-myopic Agents, EC'22 (Section 4.3), which refers back to Stochastic multiarmed bandits with unrestricted delay distributions, ICML'21?*
> > >
> > > In Section 4.3 of [HLNW22], the authors consider a stochastic bandit setting in which a sequence of *rewards* is perturbed within some ball of radius $\delta$, possibly in an *adversarial* way. In contrast, in our setting the sequence of *contexts* are perturbed within some ball of radius $\delta$, in a *strategic* way (i.e. given in Definition 2.1). As a result, while both problem settings require the learner to deal with perturbations, the tools and techniques used to obtain no-regret in the two settings differ significantly. For example, their algorithm SuccessiveEliminationDelayed relies on learning confidence bounds for a set of arms using all of the delayed feedback, while our Algorithm 1 relies on greedy estimation of the relationship between contexts and rewards, using only data which has not been strategically modified.
> > >
> > > [HLNW22]: Nika Haghtalab, Thodoris Lykouris, Sloan Nietert, Alex Wei. Learning in Stackelberg Games with Non-myopic Agents, EC 2022.

---

> > > > ### Comment · Reviewer_6Gq7 · 2023-08-15
> > > >
> > > > Thanks for the quick response. I have no further question.

---

### Official Review · Reviewer_vd1A · 2023-07-06

**Soundness:** 4 excellent
**Presentation:** 4 excellent
**Contribution:** 3 good
**Rating:** 6
**Confidence:** 4

**Summary:**

They consider the problem of online learning with apple tasting feedback where the sequence of arriving agents may strategically modify their features (contexts). They show how to achieve sublinear strategic regret compared to the best policy in the hindsight if agents were reporting truthfully.

Their main result is $\tilde{O}(\sqrt{T})$ strategic regret when the sequence of agents arriving is stochastic. Their regret bound depends exponentially on d the context dimension. They show how to mitigiate this dependency and achieve a regret bound that depends polynomially on d but with a worse dependence on T. The main idea to to use a modified version of explore-then-commit algorithm that was introduced in the prior work.

Finally, they extend their results to a sequence of agents chosen by an oblivious adversary that achieves sublinear regret however it requires an exponentially large amount of computation at each round.


**Strengths:**

I think the paper is written nicely and the results are nice. The algorithms seem to be a natural extension of the non-strategic setting. The model is interesting and the presentation is great.


**Weaknesses:**

I think the paper is written nicely and the results are nice. The algorithms seem to be a natural extension of the non-strategic setting. Since they are using the linear thresholds model, they need to shift the decision boundary to account for strategic behavior and a similar set of ideas have been studied in the previous work.


**Questions:**

It might be helpful to add a paragraph on how your bounds differ from the non-strategic case.

Prop 2.4. last line, it might be helpful for the reader to argue why the first term is at most 0.

Line 218, you can ensure the data is clean since it is strictly above the threshold?

Thm 3.3. it might be helpful to also remind the reader what c_0 is.

Another open question is to get regret bounds when the agents are picked by an adaptive adversary.

Are there any connections between the trembling hand model that you are describing and the \eps-best response model considered by [Haghtalab, Lykouris, Nietert, Wei' EC22]?

Another piece of work related to your bandit feedback could be the work by [Ahmadi, Blum, Yang' EC23] where in one of their models they propose a modification of EXP3 algorithm in the strategic setting.


**Limitations:**

yes

---

> ### Author Rebuttal · Authors · 2023-08-09
>
> Thank you for taking the time to review our work. Please find our responses to your questions below.
>
> [*It might be helpful to add a paragraph on how your bounds differ from the non-strategic case.*]
>
> We would be happy to add such a paragraph in Section 1.1 where we discuss our contributions. At a high level, our main bound (i.e., the one for stochastic contexts) has a worse dependence on the context dimension, which arises as a direct result of the agents’ ability to strategize.
>
> [*Prop 2.4. last line, it might be helpful for the reader to argue why the first term is at most 0.*]
>
> The first term is at most zero since the principal’s reward from the optimal policy when the agent strategizes is at most their optimal reward when agents do not strategize. We will add a sentence saying this in our next revision.
>
> [*Line 218, you can ensure the data is clean since it is strictly above the threshold?*]
>
> Yes, we will clarify this in the revision.
>
> [*Thm 3.3. it might be helpful to also remind the reader what c_0 is.*]
>
> $c_0$ is a lower bound on the bounded density ratio and is defined in Assumption 3.1. We will clarify this in our theorem statement.
>
> [*Another open question is to get regret bounds when the agents are picked by an adaptive adversary.*]
>
> Modifying Algorithm 3 to be based on EXP3.P instead of EXP3 would allow us to handle this setting, and we will add a comment on that after the discussion on Algorithm 3.
>
> [*Are there any connections between the trembling hand model that you are describing and the \eps-best response model considered by [Haghtalab, Lykouris, Nietert, Wei' EC22]?*]
>
> Yes, the two models are similar at a high level. In particular, HLNW22 study a Stackelberg game setting in which the follower best-responds $\epsilon$-optimally. In our trembling hand setting, the strategic agent can also be thought of as $\epsilon$-best responding, although it is important to note that an $\epsilon$-best response for the agent in our setting will cause them to only strategize more than necessary (at least for sufficiently small epsilon). We will add a discussion about this connection (and a reference to HLNW22) in the revision.
>
> [*Another piece of work related to your bandit feedback could be the work by [Ahmadi, Blum, Yang' EC23] where in one of their models they propose a modification of EXP3 algorithm in the strategic setting.*]
>
> Thanks for sharing this reference. While related in the sense that they study an online strategic learning problem, ABY23 focus on the full feedback setting, whereas our primary focus is strategic learning under apple tasting and bandit feedback. Note that when ABY23 refer to ‘bandit feedback’ in Sec. 6.1, they mean that they only see the outcome under the deployed classifier (vs all possible classifiers), while we use `bandit feedback’ to refer to the fact that we only see feedback when a positive decision is made.

---

> > ### Comment · Reviewer_vd1A · 2023-08-16
> >
> > Thank you for your detailed response. I went through them and do not have any other questions at this point.

---

### Official Review · Reviewer_qhsL · 2023-07-06

**Soundness:** 3 good
**Presentation:** 3 good
**Contribution:** 2 fair
**Rating:** 6
**Confidence:** 4

**Summary:**

This paper studies an online learning problem with one-sided (apple-tasting) feedback. At each round $t\in[T]$, an agent with a $d$-dimensional context vector $x_t$ arrives. The principal chooses a policy $\pi_t$ to map the context to binary decisions. Given $\pi_t$, the agent best responds strategically by modifying their context to $x'_t$ where $||x'_t-x_t||\leq \delta$. Then, the principal receives a one-sided, linear feedback in $x_t$. The goal is to design a policy with sub-linear strategic regret that compares the principal's reward with that of the optimal policy had agents reported their contexts truthfully. The authors propose algorithms for stochastic and adversarial settings.

**Strengths:**

- Online learning with strategic agents has many real-world applications and the results of this work could be applied to automatic decision-making processes such as lending and hiring. Therefore, the problem is very well-motivated.
- The paper is very well-written, in particular, the proof sketches provided in the paper give a very clear high-level picture of the techniques and ideas used to analyze the performance of the algorithms.
- The authors have done an excellent job comparing and contrasting their paper with prior works, it is clear how this paper contributes to this literature.

**Weaknesses:**

- While the proposed algorithms are a great first step towards solving this problem, they are sub-optimal from different aspects. First, Algorithm 1 does not make any use of unclean data and it simply discards them. While the authors have shown that using unclean data for estimation is not useful, they could be used with some confidence level to rule out some of the underperforming policies. Moreover, Algorithm 3 for the adversarial setting is practically not useful because of its exponential computational complexity.
- Based on the motivating examples that are provided in the introduction, I don't see why agents strategically modifying their context vector is necessarily a bad thing. For instance, paying bills on time and maintaining low credit utilization to increase credit scores (which is used for lending) should be incentivized (rather than discouraged).

**Questions:**

- What is the reasoning behind a bounded $\ell_2$ norm perturbation of size $\delta$ (instead of a different norm or a packing-type constraint) as the effort budget? Could you explain this in the context of a motivating application?
- Why is $r_0$ missing in the linear thresholding policy of Algorithm 3?
- Instead of considering each $d$-dimensional vector in $[0,1]^d$ (or a discretized grid of such vectors) as experts in Algorithm 3, is it possible to assume a structure on the policies and solve the resulting online structured learning problem (as it's done in the following papers)?

Koolen, Wouter M., Manfred K. Warmuth, and Jyrki Kivinen. "Hedging Structured Concepts." COLT. 2010.
Cohen, Alon, and Tamir Hazan. "Following the perturbed leader for online structured learning." International Conference on Machine Learning. PMLR, 2015.

- Algorithm 4 is mentioned multiple times in the paper, but it's not provided in the main text. It'd be great to either have the algorithm or a short description of it in the paper.
- Is it possible to relax the assumption that the variance $\sigma^2$ is known? How would such a relaxation affect the results?


--------------------------------
I've read the authors' rebuttal, thanks for addressing my questions and concerns.

**Limitations:**

The limitations are clearly discussed in the conclusion. This is a theoretical work and a discussion of potential negative societal impacts is not necessary.

---

> ### Author Rebuttal · Authors · 2023-08-09
>
> Thank you for taking the time to review our work. Please find our answers to your questions below.
>
> [*What is the reasoning behind a bounded ℓ2 norm perturbation of size $\delta$ (instead of a different norm or a packing-type constraint) as the effort budget? Could you explain this in the context of a motivating application?*]
>
> We chose to model the agent’s effort budget with respect to the l2 norm in order to allow for an easier comparison to both (1) the existing literature on strategic learning (e.g., “Learning strategy-aware linear classifiers” by Chen et al., 2020) and (2) the adversarial robustness literature (see, e.g. “Certified Adversarial Robustness via Randomized Smoothing” by Cohen et al., 2019), in which an adversary is allowed to perturb the input to a machine learning algorithm such that the perturbed input is within an l2 ball of the original input. We conjecture that our results are generalizable to the setting under which the agent’s modification is constrained by an arbitrary (known) p-norm. However, this is non-trivial because the naive reduction from the l2 norm to an arbitrary p-norm is not tight (indeed, we only know of not matching upper and lower bounds), and so just substituting the appropriate upper/lower bound may result in false positives/negatives. Finally, packing-type constraints have not been studied in the literature on strategic learning to the best of our knowledge, but we believe that this would be an interesting direction for future research.
>
> [*Why is $r_0$ missing in the linear thresholding policy of Algorithm 3?*]
>
>  $r_t(a_{t,e_t} ) = r_0$ whenever the principal takes action 0. Note that according to our apple tasting feedback whenever the principal takes action 0, they do not observe feedback. In Lines 339-340 we explain how the linear thresholding policy would change if the principal were to observe feedback whenever they chose action 0 too (i.e., bandit feedback).
>
> [*Instead of considering each $d$-dimensional vector in $[0,1]^d$ (or a discretized grid of such vectors) as experts in Algorithm 3, is it possible to assume a structure on the policies and solve the resulting online structured learning problem (as it's done in the following papers)?*]
>
> We view structured learning in strategic settings as an important direction for future work as, to the best of our knowledge, it has not yet been considered even in the full feedback setting. It may be possible to adapt Algorithm 1 to a structured policy class, if we are given access to an optimization oracle for that class. This is because Algorithm 1 computes a greedy estimate of the optimal policy class in the non-strategic setting, then shifts it to account for the strategic behavior of the agents. Using an optimization oracle, one could do something similar by first estimating the optimal policy using the optimization oracle, then modify the estimated policy to account for strategic behavior.
>
> [*Algorithm 4 is mentioned multiple times in the paper, but it's not provided in the main text. It'd be great to either have the algorithm or a short description of it in the paper.*]
>
> Thank you for this suggestion, which was brought up by other reviewers as well. We will add Algorithm 4 to the main body of the paper..
>
> [*Is it possible to relax the assumption that the variance $\sigma^2$ is known? How would such a relaxation affect the results?*]
>
> Yes, thanks for pointing this out. An upper bound on the variance should suffice, as we only require knowledge of $\sigma^2$ when setting algorithm hyperparamters. In this case the same bounds will hold, but with the upper-bound on $\sigma^2$ in place of $\sigma^2$.

---

> > ### Comment · Reviewer_qhsL · 2023-08-15
> >
> > Thanks for your detailed response to my questions. I understand your motivation for choosing bounded $\ell_2$ norm perturbations, however, I believe an $\ell_2$ ball centered around the original context is not necessarily an accurate representation of the strategic behavior of the agents. For instance, the agents might be able to change some of their contexts more easily than others (an ellipsoid rather than a sphere) or there might be different costs per unit of changes to each of their contexts (motivating packing constraints). Also, it'd be great if you commented on the second point in the "Weaknesses" section of my review (regarding why agents strategically modifying their context vector is necessarily a bad thing).

---

> > > ### Author Response · Authors · 2023-08-15
> > >
> > > [*$\ell_2$ ball vs ellipse*]
> > >
> > > Thanks for pointing this out. Our results extend readily to the setting in which the agent's effort constraint takes the form of an ellipse rather than a sphere. Under this setting, the agent effort budget constraint in Definition 2.1 would be $||A^{1/2} (x' - x_t)||_2 \leq \delta$, where $A$ is some positive definite matrix. (We note that such an effort budget is also considered in [10], although under a different setting than ours.) If the matrix $A$ is known to the principal, this can be viewed as just a linear change in the feature representation, and therefore all of our results will carry over. We will make a note of this in the revision.
> > >
> > >
> > > [*why agents strategically modifying their context vector is necessarily a bad thing*]
> > >
> > > You are correct that there are some settings in which it makes sense to consider both undesirable strategizing (i.e. "gaming") and desirable strategizing (i.e. "improvement"). Indeed, there are several works in the literature on strategic learning which consider both gaming and improvement (e.g., [9, 32]). In this work, our main goal is to study the effects of apple tasting feedback when learning under agent incentives. As a result, we chose to focus on designing policies for the principal that are robust to manipulation/gaming (a common goal in the literature, see e.g. [15, 21]). However, we view such an extension to both gaming and improvement as an interesting direction for future research.

---

> > > > ### Comment · Reviewer_qhsL · 2023-08-16
> > > >
> > > > Thanks for your response. If your work focuses on "gaming" (rather than "improvement"), the motivating example in the Introduction about “9 Ways to Build and Improve Your Credit Fast” is not applicable because paying bills on time and maintaining low credit utilization to increase credit scores (which is used for lending) should be incentivized (rather than discouraged). I encourage the authors to clarify this point in the paper and use alternative motivating applications that apply to their problem framework.

---

> > > > > ### Author Response · Authors · 2023-08-16
> > > > >
> > > > > We thank the reviewer for raising this point, and we will be sure to clarify in the revision. We agree that a reasonable goal of the principal in strategic settings such as credit lending is to both (i) discourage gaming and (ii) encourage improvement. As stated in our previous message, we chose to focus only on gaming as our main goal is to study the effects of apple tasting feedback when learning under agent incentives. Mapping this into the lending example, this would correspond to a principal which aims to be "robust" to behavior which does *not* improve one's creditworthiness (e.g. strategizing w.r.t. credit card utilization rates by opening more cards and using them all less), as opposed to actively trying to encourage behavior which does improve creditworthiness (e.g. actually reducing existing debt).

---

### Decision · Program_Chairs · 2023-09-21

**Decision:**

Accept (poster)

**Comment:**

Overall, reviewers are all leaning towards acceptance. They think the paper studies an interesting and well-motivated problem, and the paper is well-written.

The reviewers also mentioned concerns on some assumptions and the disconnection between the paper's setting and motivating examples. It might be good add more discussions for these concerns.